# Enhancing Image-Conditional Coverage in Segmentation: Adaptive Thresholding via Differentiable Miscoverage Loss

**Rui Luo**[1]*, **Jie Bao**[2], **Xiaoyi Su**[1], **Wen Jung Li**[1], **Suqun Cao**[2]
[1]City University of Hong Kong, [2]Huaiyin Institute of Technology

## Abstract

Current deep learning models for image segmentation often lack reliable uncertainty quantification, particularly at the image-specific level. While Conformal Risk Control (CRC) offers marginal statistical guarantees, achieving image-conditional coverage, which ensures prediction sets reliably capture ground truth for individual images, remains a significant challenge. This paper introduces a novel approach to address this gap by learning image-adaptive thresholds for conformal image segmentation. We first propose **AT** (Adaptive Thresholding), which frames threshold prediction as a supervised regression task. Building upon the insights from AT, we then introduce **COAT** (Conditional Optimization for Adaptive Thresholding), an innovative end-to-end differentiable framework. COAT directly optimizes image-conditional coverage by using a soft approximation of the True Positive Rate (TPR) as its loss function, enabling direct gradient-based learning of optimal image-specific thresholds. This novel differentiable miscoverage loss is key to enhancing conditional coverage. Our methods provide a robust pathway towards more trustworthy and interpretable uncertainty estimates in image segmentation, offering improved conditional guarantees crucial for safety-critical applications. The code is available at `https://github.com/bjbbbb/Conditional-Optimization-for-Adaptive-Thresholding`.

## 1 Introduction

Image segmentation is a fundamental computer vision task with critical applications in medical diagnostics, autonomous driving, and remote sensing. While deep learning has significantly advanced segmentation performance, reliable uncertainty quantification remains challenging but essential for safety-critical applications. Traditional evaluation metrics like Dice or IoU provide overall performance measures but fail to offer instance-wise reliability guarantees.

Conformal prediction (CP) has emerged as a powerful framework for providing distribution-free uncertainty quantification with finite-sample guarantees. It constructs prediction regions that contain the true label with a user-specified probability, regardless of the underlying data distribution. Recent work on Conformal Risk Control (CRC) (Angelopoulos et al., 2024) has extended this framework to handle more complex performance metrics beyond simple miscoverage, such as controlling the false negative rate in segmentation tasks.

However, a key limitation of standard CRC, particularly in image-level tasks like segmentation, is its focus on *marginal* guarantees. While CRC ensures that the average risk across a dataset is controlled, the risk for individual images (i.e., the *conditional* risk) can vary substantially. In safety-critical domains, ensuring that each image's prediction is reliable, rather than just the average over many images, is paramount. This image-specific variability in risk is a significant challenge that current approaches struggle to address effectively.

This paper proposes a novel approach to achieve image-conditional coverage in conformal image segmentation by learning image-adaptive thresholds. Our core idea is to train a model that predicts

---

*Corresponding author. E-mail: ruiluo@cityu.edu.hk

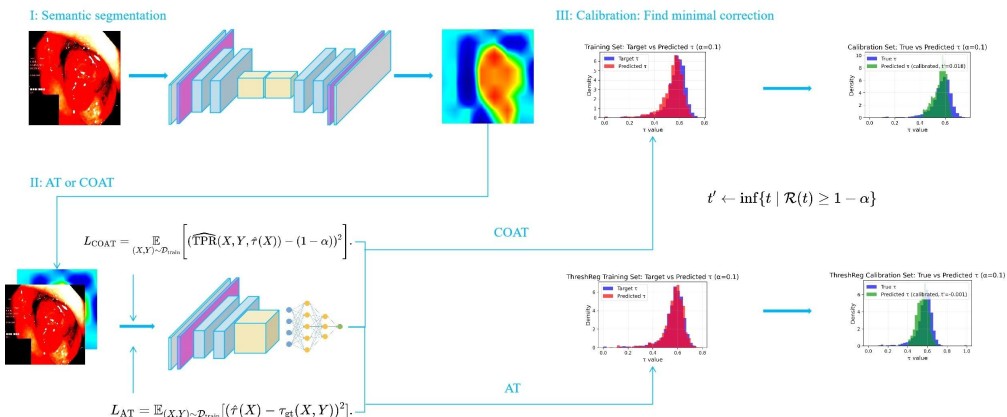

Figure 1: Schematic Overview of the COAT Framework Pipeline.

a unique threshold for each input image, aiming to satisfy the desired coverage level for that specific image. We introduce two distinct methods to realize this:

1. **AT** (Adaptive Thresholding): As an initial step, this method treats the problem of threshold prediction as a supervised regression task. We pre-compute optimal hard thresholds for training images that achieve the target coverage, and then train a neural network to predict these thresholds given the image and its base segmentation model outputs.

2. **COAT** (Conditional Optimization for Adaptive Thresholding): Building upon AT's concept, we propose an innovative end-to-end differentiable framework, which we name COAT (as illustrated in Figure 1). Instead of relying on pre-computed hard thresholds, COAT directly optimizes for image-conditional coverage. It achieves this by utilizing a soft, differentiable approximation of the True Positive Rate (TPR) to define its loss function, enabling direct gradient-based learning of optimal image-specific thresholds. This novel differentiable miscoverage loss is a key contribution for enhancing conditional coverage.

By learning image-adaptive thresholds and, particularly through the end-to-end differentiable optimization of COAT, our methods provide a robust pathway towards more trustworthy and interpretable uncertainty estimates in image segmentation, offering significantly improved conditional guarantees, which are crucial for the deployment of AI systems in high-stakes applications.

## 2 PRELIMINARIES AND PROBLEM SETUP

### 2.1 PROBLEM SETUP

For image segmentation, we consider an input image $X_i$ with its ground truth binary mask $Y_i \in \{0,1\}^N$, which delineates a region of interest. Our primary objective is to construct a prediction set $\widehat{C}(X_i) \subset \{1, 2, ..., N\}$ that controls the false negative rate (FNR) in expectation. The FNR quantifies the proportion of true positive pixels that are incorrectly excluded from the prediction set, a critical metric in applications such as medical imaging where missing regions of interest can have severe consequences. Specifically, we aim to ensure:

$$\mathbb{E}\left[1 - \frac{\sum_{j=1}^{N} \widehat{C}(X_i)[j] \cdot Y_i[j]}{\sum_{j=1}^{N} Y_i[j]}\right] \leq \alpha, \qquad (1)$$

where $\alpha \in (0, 1)$ is a user-specified risk level. Here, $\sum_{j=1}^{N} Y_i[j]$ denotes the number of positive pixels in the ground truth mask. We also define $\epsilon$ as a small positive constant (e.g., $10^{-6}$) used to prevent division by zero in certain calculations. The expectation in equation 1 is taken over random draws of the test data, reflecting the average performance of the prediction set $\widehat{C}(X_i)$.

However, this marginal guarantee, while ensuring that the average FNR across the entire dataset is controlled, does not guarantee consistent performance for individual images. Due to the inherent variability in image "difficulty" or characteristics, applying a single threshold to all images can lead to over-coverage for "easy" images and severe under-coverage for "difficult" ones. This implies that, while the FNR might be met on average, the conditional coverage (i.e., $1 - \text{FNR}$ for a single image) for specific images can deviate significantly from the target level $1 - \alpha$. For safety-critical applications, such image-to-image variability is unacceptable, necessitating a stronger guarantee: not only must the marginal FNR be controlled, but the prediction reliability for each image should also be as close as possible to the target level.

That is, for an input image $X_i \in \mathcal{D}_{\text{test}}$ and its corresponding ground-truth label $Y_i \in \mathcal{D}_{\text{test}}$, we should ensure the fulfillment of the following conditions:

$$\text{Coverage} = \frac{1}{|\mathcal{D}_{test}|} \sum_{X_i \in \mathcal{D}_{test}} \frac{\sum_{j=1}^{N} \widehat{C}(X_i)[j] \cdot Y_i[j]}{\sum_{j=1}^{N} Y_i[j]}. \tag{2}$$

Subsequently, we should strive to narrow the coverage gap (Kaur et al., 2025):

$$\text{Coverage Gap} = \frac{1}{|\mathcal{D}_{test}|} \sum_{X_i \in \mathcal{D}_{test}} \left( \left| \frac{\sum_{j=1}^{N} \widehat{C}(X_i)[j] \cdot Y_i[j]}{\sum_{j=1}^{N} Y_i[j]} - (1 - \alpha) \right| \right). \tag{3}$$

## 2.2 CONFORMAL RISK CONTROL (CRC)

Conformal Risk Control (CRC) (Angelopoulos et al., 2024) extends the principles of conformal prediction to offer distribution-free guarantees on the expected value of any monotone loss function. For image segmentation, CRC achieves control over the false negative rate by applying a specific threshold to the pixel-wise probabilities generated by a base segmentation model. This optimal threshold is determined through a calibration procedure performed on a dedicated held-out dataset.

Given a base segmentation model that outputs a probability map $\widehat{p}(X_i) = (\widehat{p}_1(X_i), \ldots, \widehat{p}_N(X_i))$ for an input image $X_i$, where $\widehat{p}_j(X_i)$ estimates $\mathbb{P}(j \in Y_i | X_i)$ for pixel $j$, the CRC approach proceeds as follows:

1. **Prediction Set Definition**: For a given threshold $\tau$, the prediction set $\widehat{C}(X_i, \tau)$ is defined by including all pixels $j$ whose predicted probability $\widehat{p}_j(X_i)$ is greater than or equal to $\tau$:

$$\widehat{C}(X_i, \tau) = \{j : \widehat{p}_j(X_i) \geq \tau\}. \tag{4}$$

2. **Calibrated Threshold Computation**: The calibrated threshold $\tau'$ is determined using a calibration dataset $D_{\text{cal}}$. It is the largest threshold that satisfies the empirical risk constraint on the calibration set:

$$\tau' = \sup\left\{ \tau : \frac{1}{n+1} \sum_{i \in \mathcal{D}_{\text{cal}}} \left( 1 - \frac{\sum_{j=1}^{N} \widehat{C}(X_i, \tau)[j] \cdot Y_i[j]}{\sum_{j=1}^{N} Y_i[j]} \right) + \frac{B}{n+1} \leq \alpha \right\}, \tag{5}$$

where $n$ is the size of the calibration set $\mathcal{D}_{\text{cal}}$, $B$ is an upper bound of the loss function (typically $B = 1$ for the FNR).

3. **Final Prediction Set**: The final prediction set for a new test image $X_i$ is then constructed using the calibrated threshold $\tau'$:

$$\widehat{C}(X_i) = \widehat{C}(X_i, \tau'). \tag{6}$$

This methodology guarantees that $\mathbb{E}\left[ 1 - \frac{\sum_{j=1}^{N} \widehat{C}(X_i)[j] \cdot Y_i[j]}{\sum_{j=1}^{N} Y_i[j]} \right] \leq \alpha$ over the data distribution, providing a distribution-free control of the false negative rate, as established by Theorem 1 in Angelopoulos *et al.* (Angelopoulos et al., 2024).

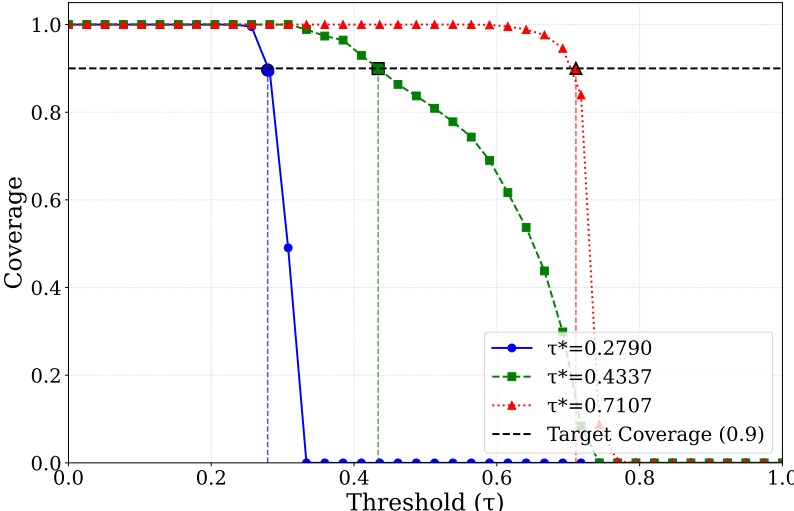

Figure 2: A figure presenting the relationship between the coverage rates of various images and the variable $\tau$.

## 3 METHODOLOGY

We introduce two novel methods for learning an image-adaptive threshold $\widehat{\tau}(X)$ for conformal risk control in image segmentation. The first, **AT** (Adaptive Thresholding), serves as a supervised baseline, while the second, **COAT** (Conditional Optimization for Adaptive Thresholding), is our more advanced, end-to-end differentiable approach.

### 3.1 AT: SUPERVISED THRESHOLD PREDICTION

The AT approach frames the problem as learning a direct mapping from an image to its optimal segmentation threshold.

#### 3.1.1 PREDICTION

The threshold predictor $f_D$ takes the input image $X$ and the corresponding probability map $\widehat{p}(X)$ from the base model to predict a single scalar threshold $\widehat{\tau}(X)$.

$$\widehat{\tau}(X) = f_D(X, \widehat{p}(X)). \tag{7}$$

The notation $(X, \widehat{p}(X))$ implies a combination of these inputs, typically through channel-wise concatenation and any necessary spatial alignment, to form the input tensor for $f_D$.

#### 3.1.2 TRAINING AND LOSS FUNCTION

This method requires a pre-computation step to generate a "ground-truth" threshold $\tau^*(X, Y)$ for each image in the training set. This $\tau^*$ is determined via numerical search (e.g., binary search) as the value that makes the TPR of the resulting hard segmentation mask equal to the target coverage level $1 - \alpha$.

The model $f_D$ is then trained using a standard Mean Squared Error (MSE) loss between the predicted threshold $\widehat{\tau}(X)$ and the ground-truth threshold $\tau^*$. The loss function over the training distribution is:

$$L_{\text{AT}} = \mathbb{E}_{(X,Y) \sim \mathcal{D}_{\text{train}}}[(\widehat{\tau}(X) - \tau^*(X, Y))^2]. \tag{8}$$

## 3.2 COAT: End-to-End Differentiable Miscoverage Loss

As shown in Figure 2, due to the non-continuous and non-increasing relationship between $\tau$ and the target coverage across different images, directly training $\tau$ may not necessarily yield satisfactory coverage performance and requires pre-calculating the relationship between $\tau$ and coverage. Moreover, the non-continuous and non-increasing nature also renders direct training of coverage infeasible. To circumvent the need for pre-calculating ground-truth thresholds, COAT enables end-to-end training by defining a fully differentiable loss function that directly optimizes for the target coverage.

### 3.2.1 Prediction

The prediction model $f_D$ has the same architecture as in AT, and it also takes the probability map from the base model as input.

$$\widehat{\tau}(X) = f_D(X, \widehat{p}(X)). \tag{9}$$

---

**Algorithm 1** COAT: Conditional Optimization for Adaptive Thresholding

---

1: **Input:** labeled data $\mathcal{D}_{\text{train}}$, unlabel test data $\mathcal{D}_{\text{test}}$, target coverage $1 - \alpha$, temperature $T$, small constant $\epsilon$.
2: Randomly split $\mathcal{D}_{\text{train}}$ into $\mathcal{D}_1$, $\mathcal{D}_2$, and $\mathcal{D}_{\text{cal}}$.
3: Train a semantic segmentation model using $\mathcal{D}_1$.
4: Initialize parameters of threshold predictor $f_D$.
5: **for** epoch in training epochs **do**
6:     **for** each $(X_i, Y_i) \in \mathcal{D}_2$ **do**
7:         Obtain probability map $\widehat{p}_i \leftarrow \widehat{p}(X_i)$.
8:         Predict threshold: $\widehat{\tau}_i \leftarrow f_D(X_i, \widehat{p}(X_i))$.
9:         Compute soft mask: $M_{\text{soft}}(X_i) = \sigma\left(\frac{\widehat{p}(X_i) - \widehat{\tau}(X_i)}{T}\right)$.
10:        Compute differentiable TPR: $\widehat{\text{TPR}}(X_i, Y_i, \widehat{\tau}(X_i)) = \frac{\sum_{j=1}^{N} M_{\text{soft}}(X_i)[j] \cdot Y_i[j]}{\sum_{j=1}^{N} Y_i[j] + \epsilon}$.
11:        Compute loss: $L_{\text{COAT}} \leftarrow (\widehat{\text{TPR}}(X_i, Y_i, \widehat{\tau}(X_i)) - (1 - \alpha))^2$.
12:        Update parameters of $f_D$ by descending the gradient $\nabla L_{\text{COAT}}$.
13:     **end for**
14: **end for**
15: Compute base thresholds: $\widehat{\tau}_i \leftarrow f_D(X_i, \widehat{p}(X_i))$ for $(X_i, Y_i, \widehat{p}(X_i)) \in \mathcal{D}_{\text{cal}}$.
16: Define coverage function: $\mathcal{R}(t) \leftarrow \frac{1}{|\mathcal{D}_{\text{cal}}|} \sum_i \frac{\sum_{j=1}^{N} \mathbb{1}[\widehat{p}_j(X_i) \geq \widehat{\tau}_i - t] \cdot Y_i[j]}{\sum_{j=1}^{N} Y_i[j]}$.
17: Find minimal correction: $t' \leftarrow \inf\{t \mid \mathcal{R}(t) \geq (|\mathcal{D}_{\text{cal}}| + 1)(1 - \alpha)/|\mathcal{D}_{\text{cal}}|\}$.
18: **for** each $(X_i, \widehat{p}(X_i)) \in \mathcal{D}_{\text{test}}$ **do**
19:     Compute the base threshold: $\widehat{\tau}_i \leftarrow f_D(X_i, \widehat{p}(X_i))$.
20:     Calculate the adjusted threshold after calibration: $\tau'_i \leftarrow \text{clip}(\widehat{\tau}_i - t', 0, 1)$.
21:     Generate the prediction set: $\widehat{C}(X_i) \leftarrow \{\widehat{p}(X_i) \geq \tau'_i\}$.
22: **end for**
23: **Output:** $\widehat{C}(X_i)$ for $i \in \mathcal{D}_{\text{test}}$.

---

### 3.2.2 Training and Loss Function

The core of this method is a differentiable approximation of the TPR. Instead of applying a hard threshold, we use the predicted threshold $\widehat{\tau}(X)$ to generate a soft, probabilistic segmentation mask $M_{\text{soft}}$. The loss is then the MSE between the TPR calculated from this soft mask and the target coverage level $1 - \alpha$.

The full loss function is defined as follows:

$$L_{\text{COAT}} = \mathop{\mathbb{E}}_{(X,Y) \sim \mathcal{D}_{\text{train}}} \left[ (\widehat{\text{TPR}}(X, Y, \widehat{\tau}(X)) - (1 - \alpha))^2 \right],$$

where the predicted TPR, $\widehat{\mathrm{TPR}}$, is computed via:

$$\widehat{\mathrm{TPR}}(X, Y, \widehat{\tau}(X)) = \frac{\sum_{j=1}^{N} M_{\mathrm{soft}}(X)[j] \cdot Y[j]}{\sum_{j=1}^{N} Y[j] + \epsilon}.$$

Here, $Y[h, w]$ denotes the pixel value at $(h, w)$ for the ground truth mask $Y$. The soft mask $M_{\mathrm{soft}}$ is defined using the sigmoid function $\sigma(\cdot)$ and a temperature parameter $T > 0$:

$$M_{\mathrm{soft}}(X) = \sigma\left(\frac{\widehat{p}(X) - \widehat{\tau}(X)}{T}\right).$$

This formulation allows gradients to flow from the final loss back to the parameters of the threshold predictor $f_D$, enabling direct optimization towards the desired conditional coverage without intermediate supervision. As detailed in Algorithm 1, we present a comprehensive description of the COAT framework for image segmentation, covering all critical implementation components.

**Remark:** The primary objective of COAT is to learn the intricate relationship between an image's characteristics and its target conditional coverage. COAT achieves this through an innovative end-to-end differentiable miscoverage loss, which directly optimizes for the desired conditional coverage. This direct optimization circumvents the need to explicitly pre-calculate the complex, non-linear relationship between individual $\tau$ and coverage for each image. Following this learning phase, a calibration set is used to apply a global adjustment, $t'$, to the predicted image-specific thresholds. This final calibration step, which can involve either a positive or negative $t'$, statistically ensures the marginal coverage rate, as defined by Equation 2, across the entire dataset. This two-stage process—directly optimizing for image-conditional reliability and then performing a marginal calibration—provides a robust pathway towards more trustworthy and interpretable uncertainty estimates.

### 3.3 THEORETICAL GUARANTEES

**Theorem 1** (Coverage Guarantees). *Let $\mathcal{D}_{cal} = \{(X_i, Y_i)\}_{i=1}^{n}$ be the calibration set and $(X_{n+1}, Y_{n+1})$ be a test sample. Suppose $\{(X_i, Y_i)\}_{i=1}^{n+1}$ are exchangeable. Then the final prediction set $\widehat{C}(X_{n+1})$ given by AT or COAT satisfies:*

$$\mathbb{E}\left[\frac{\sum_{j=1}^{N} \widehat{C}(X_{n+1})[j] \cdot Y_{n+1}[j]}{\sum_{j=1}^{N} Y_{n+1}[j]}\right] \geq 1 - \alpha. \tag{10}$$

*Proof.* For each sample $(X_i, Y_i)$, we define a parameterized loss function $L_i(t)$ for $t \in [0, 1]$, where $t$ is a global correction parameter. The loss is the false negative rate (FNR) for the prediction set formed by the adjusted threshold:

$$L_i(t) = 1 - \frac{\sum_{j=1}^{N} \mathbb{1}[\widehat{p}_j(X_i) \geq \mathrm{clip}(\widehat{\tau}_i - t, 0, 1)] \cdot Y_i[j]}{\sum_{j=1}^{N} Y_i[j]}. \tag{11}$$

Here, $\widehat{\tau}_i = f_D(X_i, \widehat{p}(X_i))$ is the base threshold predicted by AT or COAT. This loss function $L_i(t)$ is a non-increasing and right-continuous function of $t$. It satisfies

$$L_i(1) = 0 \leq \alpha \quad \text{and} \quad \sup_t L_i(t) \leq 1. \tag{12}$$

We compute the empirical risk on the calibration set:

$$\bar{L}_n(t) = \frac{1}{n}\sum_{i=1}^{n} L_i(t) = 1 - \mathcal{R}(t), \tag{13}$$

where $\mathcal{R}(t)$ is the empirical coverage defined in Algorithm 1. Define

$$t' = \inf\left\{t \,\middle|\, \frac{n}{n+1}\bar{L}_n(t) + \frac{1}{n+1} \leq \alpha\right\} = \inf\left\{t \,\middle|\, \mathcal{R}(t) \geq \frac{n+1}{n}(1 - \alpha)\right\}. \tag{14}$$

By Theorem 1 in Angelopoulos et al. (2024), we have $\mathbb{E}\left[L_{n+1}(t')\right] \leq \alpha$, which implies

$$\mathbb{E}\left[\frac{\sum_{j=1}^{N} \widehat{C}(X_{n+1})[j] \cdot Y_{n+1}[j]}{\sum_{j=1}^{N} Y_{n+1}[j]}\right] \geq 1 - \alpha, \tag{15}$$

as $\widehat{C}(X_{n+1})[j] = \mathbb{1}[\widehat{p}_j(X_{n+1}) \geq \mathrm{clip}(\widehat{\tau}_{n+1} - t', 0, 1)]$. $\qquad \square$

This theorem shows that both AT and COAT provide a powerful finite-sample guarantee for marginal coverage, i.e., the mean TPR. We also discuss in Appendix A.1 the assumptions under which they can achieve conditional validity.

## 3.4 RELATED WORK

### 3.4.1 CONFORMAL RISK CONTROL

Unlike other uncertainty quantification methods (Chen et al., 2025a;c;b), conformal prediction (Vovk et al., 2005) provides distribution-free uncertainty quantification with finite-sample guarantees. Its split conformal variant (Lei et al., 2018) is widely used for its computational efficiency. CP has been successfully applied to both classification (Luo & Colombo, 2024; Luo & Zhou, 2024) and regression tasks (Luo & Zhou, 2025c;d; Bao et al., 2025a; Guo et al., 2026). Its flexibility allows adaptation to various real-world scenarios, including games (Bao et al., 2025b; Luo et al., 2026), time-series forecasting (Su et al., 2024), and graph-based applications (Luo et al., 2023; Tang et al., 2025; Luo & Zhou, 2025b; Wang et al., 2025a;b; Luo & Colombo, 2025; Zhang et al., 2025). Recent extensions, notably Conformal Risk Control (CRC) (Angelopoulos et al., 2024; Bates et al., 2021), allow for controlling the expected value of various monotone loss functions, such as the false negative rate in segmentation.

While initial CRC applications often used a single global threshold, recent works have explored more nuanced control. Teneggi *et al.* (Teneggi et al., 2023) proposed grouping pixels to control risk through a convex surrogate loss, and further extended this to semantic-specific control for medical imaging (Teneggi et al., 2025). Bereska *et al.* (Bereska et al., 2025) introduced Spatially-Aware Conformal Prediction (SACP) to adapt uncertainty estimates based on proximity to critical structures. He *et al.* (He et al., 2025) integrated CRC into model training for quality assurance in radiation therapy. These advancements highlight a growing need for more adaptive and context-aware risk control beyond global guarantees, which our image-conditional approach directly addresses.

### 3.4.2 CONDITIONAL CONFORMAL PREDICTION

Achieving conditional coverage – ensuring prediction sets attain the desired coverage for every possible covariate value – is generally impossible without strong distributional assumptions (Vovk, 2012; Foygel Barber et al., 2021). However, for high-stakes applications, marginal guarantees are often insufficient due to potential disparities in coverage across subpopulations.

Many works aim to improve conditional validity by modifying the calibration step (Lei & Wasserman, 2014; Guan, 2023; Barber et al., 2023; Luo & Zhou, 2025a) or altering the initial prediction rule (Romano et al., 2019; Sesia & Romano, 2021; Chernozhukov et al., 2021). Some research focuses on coverage under covariate shift (Lei & Wasserman, 2014; Tibshirani et al., 2019; Izbicki et al., 2022), with frameworks like that by Gibbs *et al.* (Gibbs et al., 2025) aiming for exact finite-sample coverage across shifts.

Group conditional guarantees have also been explored (Toccaceli & Gammerman, 2019; Gupta et al., 2020; Ding et al., 2023). Mondrian conformal prediction (Vovk et al., 2003) provides exact coverage for disjoint groups. More flexible methods for overlapping groups exist (Foygel Barber et al., 2021; Jung et al., 2023), though they can be computationally intensive or rely on distributional assumptions. Other approaches learn features to improve conditional coverage, such as density-based atypicality (Yuksekgonul et al., 2023) or learning covariate space partitions (Kiyani et al., 2024).

Our method contributes to this line by providing a novel way to achieve image-conditional coverage in segmentation, moving beyond group-based approaches to an instance-specific adaptive thresholding mechanism, particularly through the end-to-end differentiable optimization.

## 4 EXPERIMENTS

This section presents the empirical evaluation of our proposed AT and COAT methods. We assess their performance in controlling the False Negative Rate (FNR) on diverse image segmentation tasks. Our experiments aim to demonstrate the effectiveness of image-adaptive thresholding compared to approaches like CRC Angelopoulos et al. (2024) and AA-CRC (Blot et al., 2025), and to validate

the robustness of our methods across different base segmentation models and datasets. Detailed experimental settings can be found in Appendixs A.2 and A.3.

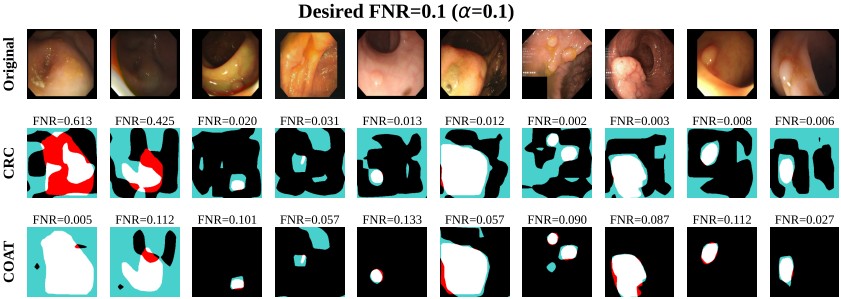

Figure 3: Qualitative comparison of CRC and COAT prediction sets at a significance level of $\alpha = 0.1$. The top row shows original polyp images, the middle row displays CRC prediction sets, and the bottom row presents COAT prediction sets. White pixels represent true positives, red false negatives, and cyan false positives. FNR values are given for each prediction. COAT demonstrates more consistent coverage and false negatives across images compared to CRC, highlighting the merit of our conditional risk control approach.

As shown in Figure 3 and Figure 4, we conducted a qualitative analysis on the polyp dataset (with alpha = 0.1) and the skin dataset (with alpha = 0.2) respectively. The base segmentation models presented in both cases are PSPNet. It can be observed that COAT is capable of better maintaining the given target coverage rate for each image. On the polyp dataset, COAT maintains a more stable FNR while also achieving a lower false positive rate. On the skin dataset, COAT consistently keeps the images at the given coverage rate. COAT adaptively adjusts thresholds to achieve the target False Negative Rate (FNR); while this may increase false positives, such instances often stem from the base model's inherent uncertainties. In safety-critical scenarios where false negatives are more detrimental than false positives, COAT's precise FNR control proves a significant advantage for robust risk management.

As shown in Table 1 and Figure 6, we randomly partitioned the dataset 20 times and then tested the mean and standard deviation of Marginal Coverage and Coverage Gap. Across all base models and datasets, COAT consistently outperformed CRC and AA-CRC in terms of Coverage Gap, with COAT consistently achieving the best Coverage Gap.

In addition, we have plotted the training progress of COAT. As can be seen from the loss function in Figure 5, regardless of the segmentation model employed and the corresponding dataset used, the training of COAT's loss function is highly stable, with a rapid decline that eventually approaches zero. The qualitative results for the Fire dataset are in Appendix A.4. We also analyzed the reference temperature $T$ in the COAT method, examining how different temperatures affect the coverage gap (see Appendix A.5). We also discussed other aspects such as the time consumption of these methods (see Appendices A.6-A.10).

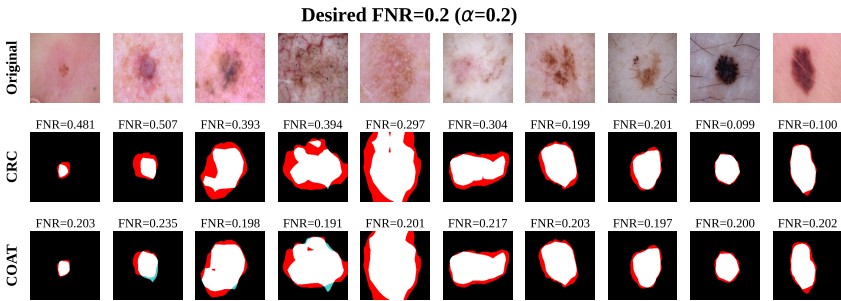

Figure 4: Qualitative comparison of CRC and COAT prediction sets at a significance level of $\alpha = 0.2$.

| Dataset | Model | Method | $\alpha = 0.1$ | | $\alpha = 0.2$ | |
| --- | --- | --- | --- | --- | --- | --- |
| | | | Marginal Coverage | Coverage Gap | Marginal Coverage | Coverage Gap |
| Polyp | Deeplab v3+ | CRC | 0.907 (0.015) | 0.145 (0.010) | 0.796 (0.028) | 0.232 (0.007) |
| | | AA-CRC | 0.900 (0.017) | 0.122 (0.015) | 0.797 (0.025) | 0.167 (0.020) |
| | | AT | 0.899 (0.024) | 0.115 (0.014) | 0.802 (0.047) | 0.174 (0.019) |
| | | COAT | 0.899 (0.016) | **0.106 (0.010)** | 0.802 (0.020) | **0.162 (0.011)** |
| | UNet | CRC | 0.901 (0.018) | 0.135 (0.013) | 0.797 (0.025) | 0.256 (0.016) |
| | | AA-CRC | 0.908 (0.017) | 0.131 (0.012) | 0.793 (0.024) | 0.217 (0.017) |
| | | AT | 0.903 (0.013) | 0.127 (0.009) | 0.805 (0.022) | 0.209 (0.013) |
| | | COAT | 0.904 (0.022) | **0.122 (0.012)** | 0.803 (0.018) | **0.199 (0.014)** |
| | PSPNet | CRC | 0.906 (0.019) | 0.150 (0.015) | 0.804 (0.026) | 0.249 (0.008) |
| | | AA-CRC | 0.908 (0.018) | 0.119 (0.016) | 0.796 (0.022) | 0.162 (0.025) |
| | | AT | 0.899 (0.018) | 0.119 (0.014) | 0.796 (0.020) | 0.166 (0.014) |
| | | COAT | 0.894 (0.016) | **0.110 (0.015)** | 0.796 (0.021) | **0.144 (0.013)** |
| | SINet | CRC | 0.904 (0.018) | 0.149 (0.014) | 0.803 (0.026) | 0.255 (0.009) |
| | | AA-CRC | 0.906 (0.026) | 0.126 (0.014) | 0.799 (0.022) | 0.182 (0.014) |
| | | AT | 0.899 (0.024) | 0.119 (0.013) | 0.809 (0.051) | 0.170 (0.014) |
| | | COAT | 0.896 (0.016) | **0.102 (0.010)** | 0.800 (0.021) | **0.148 (0.014)** |
| Fire | Deeplab v3+ | CRC | 0.901 (0.003) | 0.067 (0.001) | 0.803 (0.005) | 0.092 (0.001) |
| | | AA-CRC | 0.903 (0.004) | 0.062 (0.002) | 0.804 (0.005) | 0.083 (0.006) |
| | | AT | 0.901 (0.003) | 0.061 (0.003) | 0.806 (0.031) | 0.086 (0.015) |
| | | COAT | 0.900 (0.002) | **0.058 (0.001)** | 0.799 (0.003) | **0.076 (0.002)** |
| | UNet | CRC | 0.899 (0.005) | 0.077 (0.001) | 0.802 (0.006) | 0.103 (0.001) |
| | | AA-CRC | 0.902 (0.005) | 0.068 (0.004) | 0.803 (0.005) | 0.088 (0.006) |
| | | AT | 0.900 (0.003) | 0.063 (0.003) | 0.800 (0.004) | 0.085 (0.006) |
| | | COAT | 0.900 (0.003) | **0.061 (0.001)** | 0.800 (0.003) | **0.079 (0.002)** |
| | PSPNet | CRC | 0.901 (0.003) | 0.065 (0.001) | 0.801 (0.005) | 0.091 (0.001) |
| | | AA-CRC | 0.904 (0.005) | 0.063 (0.003) | 0.808 (0.008) | 0.079 (0.010) |
| | | AT | 0.904 (0.019) | 0.065 (0.003) | 0.799 (0.004) | 0.089 (0.005) |
| | | COAT | 0.900 (0.003) | **0.060 (0.001)** | 0.800 (0.004) | **0.077 (0.002)** |
| | SINet | CRC | 0.901 (0.004) | 0.071 (0.001) | 0.800 (0.006) | 0.101 (0.001) |
| | | AA-CRC | 0.903 (0.008) | 0.063 (0.002) | 0.803 (0.009) | 0.082 (0.006) |
| | | AT | 0.900 (0.003) | 0.065 (0.004) | 0.799 (0.005) | 0.090 (0.007) |
| | | COAT | 0.900 (0.002) | **0.059 (0.002)** | 0.799 (0.004) | **0.080 (0.003)** |
| Skin | Deeplab v3+ | CRC | 0.900 (0.003) | 0.072 (0.001) | 0.802 (0.006) | 0.107 (0.002) |
| | | AA-CRC | 0.905 (0.004) | 0.057 (0.003) | 0.806 (0.005) | 0.079 (0.010) |
| | | AT | 0.904 (0.016) | 0.061 (0.009) | 0.809 (0.039) | 0.090 (0.023) |
| | | COAT | 0.899 (0.003) | **0.054 (0.001)** | 0.800 (0.005) | **0.073 (0.002)** |
| | UNet | CRC | 0.900 (0.003) | 0.062 (0.001) | 0.800 (0.006) | 0.097 (0.002) |
| | | AA-CRC | 0.908 (0.003) | 0.056 (0.003) | 0.807 (0.005) | 0.081 (0.004) |
| | | AT | 0.899 (0.003) | 0.059 (0.002) | 0.800 (0.004) | 0.090 (0.006) |
| | | COAT | 0.899 (0.003) | **0.054 (0.001)** | 0.800 (0.004) | **0.079 (0.002)** |
| | PSPNet | CRC | 0.902 (0.003) | 0.069 (0.001) | 0.804 (0.006) | 0.103 (0.001) |
| | | AA-CRC | 0.906 (0.005) | 0.057 (0.005) | 0.806 (0.005) | 0.071 (0.011) |
| | | AT | 0.903 (0.015) | 0.061 (0.008) | 0.809 (0.039) | 0.076 (0.025) |
| | | COAT | 0.899 (0.003) | **0.050 (0.002)** | 0.799 (0.004) | **0.064 (0.002)** |
| | SINet | CRC | 0.905 (0.004) | 0.078 (0.001) | 0.806 (0.004) | 0.113 (0.001) |
| | | AA-CRC | 0.905 (0.005) | 0.063 (0.006) | 0.805 (0.005) | 0.075 (0.010) |
| | | AT | 0.906 (0.022) | 0.065 (0.010) | 0.800 (0.003) | 0.074 (0.005) |
| | | COAT | 0.899 (0.003) | **0.055 (0.001)** | 0.800 (0.004) | **0.071 (0.002)** |

Table 1: Marginal Coverage and Coverage Gap Results at $\alpha = 0.1$ and $\alpha = 0.2$ Across Different Models and Conformal Methods. Each dataset result is the mean and standard deviation of 20 random splits.

**Remark on Coverage Gap Limitations:** It is important to note that the coverage gap cannot be reduced to zero in practice due to several fundamental limitations. First, finite sample effects mean that with limited calibration data, perfect estimation of image-specific coverage is statistically impossible. Second, there exists an inherent bias-variance trade-off in threshold prediction - while our adaptive methods reduce bias by learning image-specific patterns, they introduce variance through the learned predictor. Third, model capacity constraints limit how well our threshold predictor can capture the complex relationship between image characteristics and optimal thresholds.

Despite these theoretical limitations, our COAT method consistently achieves the smallest coverage gap across all experimental settings, demonstrating its effectiveness in learning meaningful image-adaptive patterns. The end-to-end differentiable optimization in COAT provides a princi-

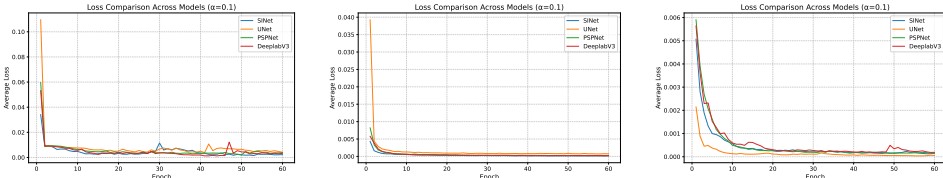

Figure 5: Loss function graphs for different segmentation models trained using the COAT method, with datasets shown from left to right being polyp, fire, and skin respectively.

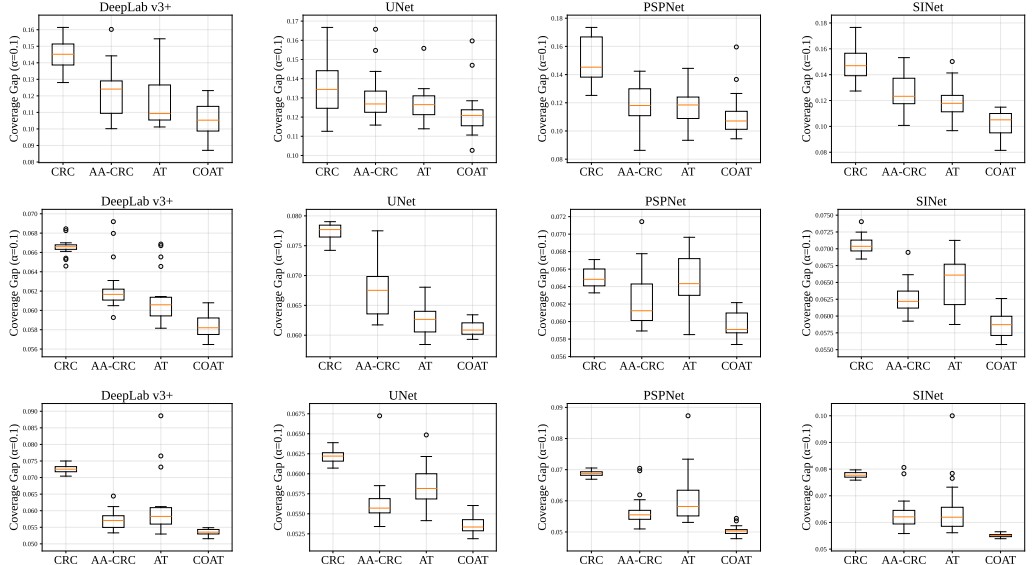

Figure 6: The box plot results for the Coverage Gap obtained from different datasets and various base segmentation models are presented. For each experimental setup, 20 random splits were conducted. The three rows of plots, from top to bottom, display the results for the polyp, fire, and skin datasets, respectively.

pled approach that directly optimizes for the target coverage, leading to more reliable conditional guarantees compared to both global thresholding (CRC) and supervised approaches (AA-CRC and AT). This improvement is particularly valuable for safety-critical applications where consistent per-image reliability is paramount.

## 5    CONCLUSION

In this paper, we address the limitation of existing segmentation methods that provide only marginal guarantees and lack image-level conditional coverage in safety-critical applications by proposing two novel methods for learning adaptive thresholds: AT and COAT. As our core contribution, COAT introduces an innovative end-to-end differentiable miscoverage loss, enabling the precise learning of an optimal threshold for each image by directly optimizing the conditional coverage target. Extensive experiments across multiple datasets demonstrate that our methods, particularly COAT, significantly reduce the Coverage Gap while maintaining the target marginal coverage rate, thereby exhibiting stronger consistency and reliability across different images. However, COAT exhibits limitations in few-shot samples, multi-class segmentation, and False Discovery Rate (FDR) risk control. This work provides a robust pathway toward building more trustworthy and interpretable AI systems for critical applications.

## ACKNOWLEDGMENTS

This work was supported by the National Natural Science Foundation of China (Grant 62506315) and City University of Hong Kong (Grants 9610639, 7020161).

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

## A APPENDIX

### LARGE LANGUAGE MODEL (LLM) USAGE

During the preparation of this manuscript, a large language model (LLM), specifically [Gemini-2.5-Flash], was employed as a general-purpose assist tool. The LLM's contributions were primarily in the following areas:

- **Code Optimization for Experimental Visualization**: The LLM assisted in optimizing and refining Python code snippets used for experimental visualization and data processing routines. This collaboration led to more efficient and readable implementations, particularly for generating the figures (e.g., Figure 3, Figure 4, Figure 7) and tables (e.g., Table 1, Table 2) presented in the paper.

- **Writing Assistance and Refinement**: The LLM was utilized for drafting and refining certain sections of the paper, including improving clarity, grammar, and stylistic coherence. This involved generating initial textual descriptions and polishing existing content to enhance its overall quality and readability.

The authors maintained full responsibility for reviewing, editing, and validating all content generated or optimized with the assistance of the LLM, ensuring its accuracy, originality, and adherence to scientific standards. The LLM was not involved in the core research ideation, experimental design, data collection, or primary analysis leading to the scientific conclusions. The scientific content, conclusions, and any potential errors remain solely the responsibility of the authors.

## A.1 Conditional Validity under Conservative Dominance

In general, distribution-free *conditional* validity is impossible without additional assumptions. Here we show that, under a conservative dominance condition (relative to an oracle conditional threshold), the adjusted thresholding rules of the proposed AT and COAT enjoy conditional validity. Let $(X, Y) \sim P$ denote a new test image–mask pair drawn from an unknown distribution on $\mathcal{X} \times \mathcal{Y}$, and let $P_X$ be the marginal distribution of $X$. Let $\widehat{\tau}(\cdot)$ be the fixed base threshold estimated by AT or COAT and $\widehat{p}(\cdot)$ be the fixed base segmentation model. For a threshold $\tau \in [0, 1]$, we define the prediction set $\widehat{C}(X, \tau) = \{j : \widehat{p}_j(X) \geq \tau\}$ and the FNR loss for $(X, Y)$:

$$L(X, Y; \tau) = 1 - \frac{\sum_{j=1}^{N} \mathbb{1}[\widehat{p}_j(X) \geq \tau] \cdot Y[j]}{\sum_{j=1}^{N} Y[j]} \in [0, 1], \tag{16}$$

where we assume $\mathbb{P}(\sum_{j=1}^{N} Y[j] > 0) = 1$. For any fixed $(x, y)$ and any $\tau \leq \tau'$, since $\widehat{C}(x, \tau) \supseteq \widehat{C}(x, \tau')$, we have

$$\sum_{j=1}^{N} \mathbb{1}[\widehat{p}_j(x) \geq \tau] \cdot y[j] \geq \sum_{j=1}^{N} \mathbb{1}[\widehat{p}_j(x) \geq \tau'] \cdot y[j], \tag{17}$$

which implies that $L(x, y; \tau) \leq L(x, y; \tau')$. Consequently, the conditional risk

$$\tilde{L}(x, \tau) := \mathbb{E}[L(X, Y; \tau) \mid X = x] = \mathbb{E}\left[1 - \frac{\sum_{j=1}^{N} \mathbb{1}[\widehat{p}_j(X) \geq \tau] \cdot Y[j]}{\sum_{j=1}^{N} Y[j]} \,\middle|\, X = x\right], \tag{18}$$

is also non-decreasing in $\tau$.

**Assumption 1** (Right-continuity). *For $P_X$-almost every $x$, the map $\tau \mapsto \tilde{L}(x, \tau)$ is right-continuous on $[0, 1]$.*

For any fixed $\alpha \in (0, 1)$ and $X = x$, we define the oracle threshold as

$$\tau^{(\alpha)}(x) := \sup\left\{\tau \in [0, 1] : \tilde{L}(x, \tau) \leq \alpha\right\}. \tag{19}$$

Under Assumption 1, we have $\tilde{L}(x, \tau^{(\alpha)}(x)) \leq \alpha$ for $P_X$-almost every $x$.

**Assumption 2** (Conservative dominance). *Fix a global correction parameter $t' \in [0, 1]$ and define the adjusted threshold $\widehat{\tau}_{t'}(x) := \mathrm{clip}(\widehat{\tau}(x) - t', 0, 1)$. Assume that*

$$\widehat{\tau}_{t'}(x) \leq \tau^{(\alpha)}(x) \quad \text{for } P_X\text{-almost every } x. \tag{20}$$

**Proposition 1** (Conditional Validity). *Under Assumptions 1 and 2, for $P_X$-almost every $x$,*

$$\mathbb{E}\left[\frac{\sum_{j=1}^{N} \mathbb{1}[\widehat{p}_j(X) \geq \widehat{\tau}_{t'}(X)] \cdot Y[j]}{\sum_{j=1}^{N} Y[j]} \,\middle|\, X = x\right] \geq 1 - \alpha. \tag{21}$$

*Proof.* Fix $x \in \mathcal{X}$. By monotonicity of $\tilde{L}(x, \tau)$ and Assumptions 1 and 2,

$$\tilde{L}(x, \widehat{\tau}_{t'}(x)) \leq \tilde{L}(x, \tau^{(\alpha)}(x)) \leq \alpha. \tag{22}$$

Equivalently,

$$\mathbb{E}\left[\frac{\sum_{j=1}^{N} \mathbb{1}[\widehat{p}_j(X) \geq \widehat{\tau}_{t'}(X)] \cdot Y[j]}{\sum_{j=1}^{N} Y[j]} \,\middle|\, X = x\right] = 1 - \tilde{L}(x, \widehat{\tau}_{t'}(x)) \geq 1 - \alpha, \tag{23}$$

for $P_X$-almost every $x$. $\qquad\square$

## A.2 DATASETS

We utilized three distinct image segmentation datasets to evaluate the robustness of our proposed algorithms: polyp segmentation, skin lesion segmentation, and flame segmentation. These tasks are particularly critical for FNR control, as missing parts of the region of interest can lead to severe consequences.

For each dataset, we followed a specific data partitioning strategy. The initial training set was used for training the base segmentation models (UNet (Ronneberger et al., 2015), DeepLab v3+ (Chen et al., 2018), PSPNet (Zhao et al., 2017), SINet (Fan et al., 2020a)). The remaining data was designated as the test set. This test set was then further partitioned. One half of the test set was reserved for final performance evaluation. The other half was designated as a calibration set. For AA-CRC, AT and COAT methods, this calibration set was further equally divided into a training subset (for training the adaptive threshold prediction model $f_D$) and a calibration subset (for determining the final calibrated threshold). For the standard Conformal Risk Control (CRC) method, the entire calibration set was utilized for its calibration procedure.

- **Polyp Dataset**: Following similar setups as Angelopoulos *et al.* (Angelopoulos et al., 2024), blot *et al.* (Blot et al., 2025) and Fan *et al.* (Fan et al., 2020b), this dataset (Jha et al., 2019; Bernal et al., 2017; Vázquez et al., 2017; Tajbakhsh et al., 2015; Silva et al., 2014) comprised 1450 images for training the base segmentation models and 798 images for the test set.
- **Skin Lesion Dataset**: We employed the HAM10000 skin image dataset (Tschandl et al., 2018). This dataset was split with 50% (5007 images) allocated for training the base models and the remaining 50% (5008 images) for the test set.
- **Fire Dataset**: For image fire segmentation experiments, we used the dataset provided by Aktaş (Aktaş, 2023). This dataset was partitioned with 80% (21968 images) for training the base models and 20% (5492 images) for the test set.

Additionally, for both the AT and COAT methods, the threshold predictor $f_D$ was implemented using a ResNet-50 (He et al., 2016) architecture. The training epochs for AT were set to 30, while for COAT, they were set to 60. For the COAT method, the temperature parameter $T$ used in the sigmoid function for the soft mask calculation was set to 0.05.

## A.3 EXPERIMENTAL DETAILS

**Reproducibility Statement**: The source code and experiment scripts used to generate the results in this paper will be made publicly available upon publication of the paper.

**Implementation Details**: All experiments were conducted on a server equipped with an NVIDIA RTX 4090 GPU (24GB of RAM), running Ubuntu 24.04. Our models were implemented using Python 3.10, PyTorch 2.3.0, and the system was configured with CUDA 12.6. The learning rate of the COAT method is consistently set to $5e^{-4}$, and the batch size is 64. All the basic segmentation models underwent 20 epochs of training. Additionally, a unified learning rate of $1e^{-4}$ and a batch size of 24 were employed for all models. The random seed was set to 42 for all experiments.

## A.4 FIRE RESULTS

Due to space constraints, we have included the qualitative analysis of the fire dataset in the appendix. It is evident that COAT consistently maintains target coverage, albeit with a certain degree of false positive rate. However, when compared to CRC, COAT demonstrates a lower and more stable false negative rate.

## A.5 SENSITIVITY ANALYSIS

In this section, we employed the SINet basic segmentation model on various datasets and conducted a grid search sensitivity analysis with parameters set as $\alpha = 0.1$ and $T = [0.001, 0.01, 0.05, 0.1, 1, 10, 100]$. Similarly, each experiment was carried out with 20 random splits, and the results are presented as the mean values and standard deviations.

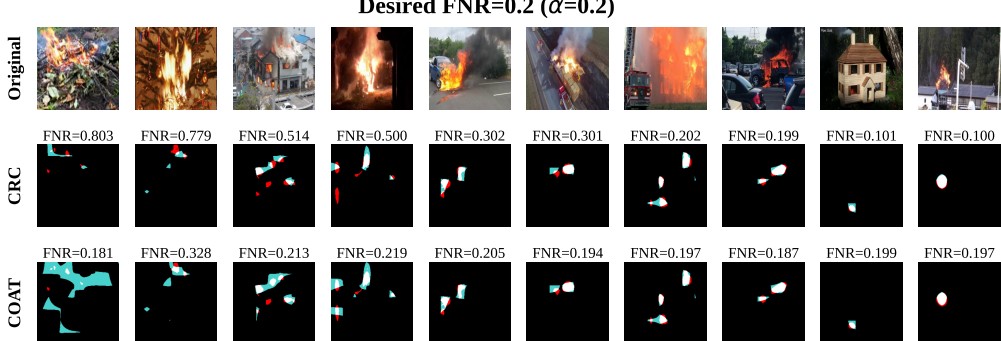

Figure 7: Qualitative comparison of CRC and COAT prediction sets at a significance level of $\alpha = 0.2$.

| Dataset | T | $\alpha = 0.1$ | |
| --- | --- | --- | --- |
| | | Marginal Coverage | Coverage Gap |
| Polyp | 100 | 0.901 (0.016) | 0.151 (0.012) |
| | 10 | 0.899 (0.023) | 0.153 (0.018) |
| | 1 | 0.899 (0.023) | 0.156 (0.017) |
| | 0.1 | 0.899 (0.012) | 0.114 (0.015) |
| | 0.05 | 0.896 (0.016) | **0.102 (0.010)** |
| | 0.01 | 0.900 (0.020) | 0.148 (0.013) |
| | 0.001 | 0.901 (0.026) | 0.147 (0.018) |
| Fire | 100 | 0.900 (0.003) | 0.071 (0.001) |
| | 10 | 0.901 (0.003) | 0.070 (0.001) |
| | 1 | 0.900 (0.003) | 0.070 (0.001) |
| | 0.1 | 0.900 (0.003) | 0.062 (0.001) |
| | 0.05 | 0.900 (0.003) | **0.059 (0.001)** |
| | 0.01 | 0.900 (0.004) | 0.060 (0.001) |
| | 0.001 | 0.900 (0.003) | 0.066 (0.014) |
| Skin | 100 | 0.900 (0.001) | 0.081 (0.001) |
| | 10 | 0.901 (0.001) | 0.081 (0.001) |
| | 1 | 0.900 (0.002) | 0.081 (0.001) |
| | 0.1 | 0.901 (0.004) | 0.056 (0.001) |
| | 0.05 | 0.899 (0.003) | **0.055 (0.001)** |
| | 0.01 | 0.900 (0.004) | 0.056 (0.002) |
| | 0.001 | 0.899 (0.003) | 0.058 (0.009) |

Table 2: Marginal Coverage and Coverage Gap Results at $\alpha = 0.1$ Across Different Models and $T$. Each dataset result is the mean and standard deviation of 20 random splits.

## A.6 TIME METRICS

In this section, we calculated the training, calibration, and testing time for each method. Specifically, we ran each method 10 times using the PSPNet segmentation model with $\alpha = 0.2$, and then computed the mean and standard deviation of the results for all methods.

## A.7 DISTRIBUTION OF CONDITIONAL COVERAGE

To further elucidate COAT's advantages, we also visualize the distribution of individual image coverage for each dataset.

Figures 8 to 10 illustrate the density distribution of coverage, for individual test images across CRC, AA-CRC, AT, and COAT. These results are obtained using the PSPNet base model on various

| Dataset | Method | Number of set samples | | | Training time (s) | Calibration time (s) | Testing time (s) |
| | | training | calibration | testing | | | |
|---|---|---|---|---|---|---|---|
| Polyp | CRC | 0 | 399 | 399 | 0.0000 (0.0000) | 0.1961 (0.0021) | 0.5414 (0.0092) |
| | AA-CRC | 200 | 198 | 399 | 38.2870 (0.3807) | 120.8917 (60.4003) | 0.4720 (0.0699) |
| | AT | 200 | 198 | 399 | 151.7819 (0.6079) | 2.1570 (0.0140) | 3.0021 (0.0631) |
| | COAT | 200 | 198 | 399 | 24.3318 (0.5924) | 0.3764 (0.0821) | 0.4807 (0.0607) |
| Fire | CRC | 0 | 2746 | 2746 | 0.0000 (0.0000) | 0.1961 (0.0021) | 0.5413 (0.0092) |
| | AA-CRC | 1373 | 1373 | 2746 | 260.0201 (0.4249) | 647.3359 (213.7731) | 3.0068 (0.0583) |
| | AT | 1373 | 1373 | 2746 | 547.3396 (0.4609) | 6.6958 (0.2020) | 9.5174 (0.1679) |
| | COAT | 1373 | 1373 | 2746 | 166.8045 (0.3589) | 2.3913 (0.0361) | 3.3785 (0.2682) |
| Skin | CRC | 0 | 2504 | 2504 | 0.0000 (0.0000) | 1.3292 (0.0315) | 0.4895 (0.0061) |
| | AA-CRC | 1252 | 1252 | 2504 | 238.7810 (0.2400) | 479.8890 (214.2507) | 2.7394 (0.03683) |
| | AT | 1252 | 1252 | 2504 | 498.5569 (1.0257) | 6.0459 (0.0214) | 8.6309 (0.0247) |
| | COAT | 1252 | 1252 | 2504 | 151.7819 (0.6079) | 2.1570 (0.0140) | 3.0021 (0.0631) |

Table 3: The time consumed by various methods during training, calibration, and testing on different datasets was recorded, utilizing the PSPNet base model with $\alpha = 0.2$ and 10 random runs.

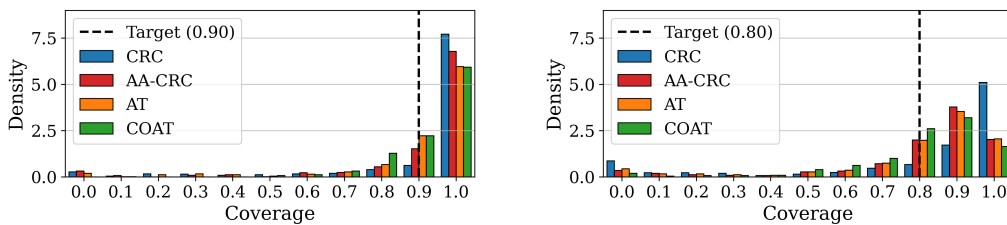

Figure 8: Density distribution of image-level coverage for CRC, AA-CRC, AT, and COAT using the Deeplab v3+ model on Polyp.

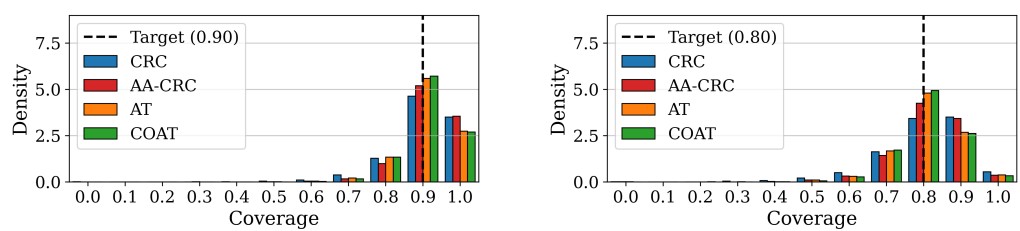

Figure 9: Density distribution of image-level coverage for CRC, AA-CRC, AT, and COAT using the Deeplab v3+ model on Fire.

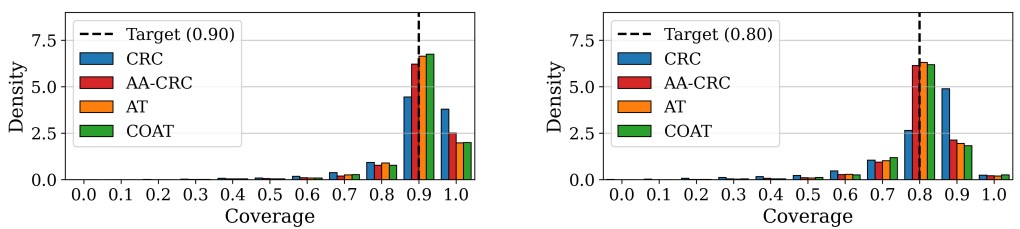

Figure 10: Density distribution of image-level coverage for CRC, AA-CRC, AT, and COAT using the Deeplab v3+ model on Skin.

datasets. The x-axis denotes the coverage per image, discretized into 0.1 intervals, while the y-axis indicates the density of images falling within each respective coverage interval. A vertical dashed line marks the target coverage of $1 - \alpha$.

A clear observation from these distributions is that COAT consistently exhibits the highest density of samples precisely at the target coverage, with a near-absence of samples in the low-coverage regions. This demonstrates that COAT provides the most consistent and reliable image-wise conditional coverage across all evaluated methods. Empirical evidence further suggests that its end-to-end differentiable optimization framework is highly effective in minimizing the discrepancy between the realized and target coverage for individual images. This, in turn, significantly enhances the trustworthiness and interpretability of uncertainty estimation in critical segmentation tasks.

## A.8 FNR and FPR Control Performance

### A.8.1 FPR Control Performance

In this section, we evaluate the performance of our COAT (Conditional Optimization for Adaptive Thresholding) method against the conventional Conformal Risk Control (CRC) approach in achieving False Positive Rate (FPR) control for image segmentation. Controlling FPR is crucial in many applications, especially when minimizing false alarms or over-segmentation is critical.

We assess performance using two key metrics:

- **Marginal Coverage:** Defined as $1 - \text{FPR}$, this metric quantifies the overall proportion of true negative pixels (background) that are correctly classified as background across the entire test set. The goal is to achieve marginal coverage close to $1 - \alpha$. For instance, if the target FPR $\alpha = 0.1$, the desired marginal coverage is $0.9$. This means we aim for the overall FPR to be at most $\alpha$.

- **Coverage Gap:** This metric measures the deviation of the conditional FPR from the target $\alpha$ for individual images. A smaller coverage gap indicates better conditional coverage, meaning the method can adaptively adjust its threshold to maintain the FPR close to $\alpha$ for each specific input image, rather than just on average. Specifically, it often reflects how well $\text{FPR}(X_i)$ adheres to $\alpha$ for any given image $X_i$. A common definition for coverage gap is $\mathbb{E}[|\text{FPR}(X_i) - \alpha|]$, or a similar measure of conditional FPR violation, where a smaller value is highly desirable.

The results in Table 4 demonstrate the superior performance of COAT in achieving robust FPR control, particularly in terms of conditional coverage. While both CRC and COAT can provide marginal FPR coverage guarantees, COAT significantly outperforms CRC by providing a much tighter conditional coverage, as evidenced by the consistently smaller coverage gaps. This ability to adapt thresholds on a per-image basis makes COAT a more reliable method for applications requiring precise FPR control and trustworthy uncertainty estimates, especially where minimizing false positives is paramount.

### A.8.2 Weighted COAT training loss control

In this section, we employ PSPNet as the base model. We prioritize the FNR loss and conduct weighted training on the FPR loss, while maintaining the calibration steps for FNR. By comparing different loss weights, we observe how the FPR changes under the condition that COAT ensures a marginal FNR.

As shown in Table 5 and Table 6, when we employ the weighted loss, we find that $\alpha = 0.1$ particularly good performance in the polyp dataset. There is a solid reason for this. In other datasets, there is an overlap between the False Negative Rate (FNR) and the False Positive Rate (FPR), which causes this part of the loss to be almost zero. However, in the polyp dataset with $\alpha = 0.1$, there is no such overlap, resulting in a relatively large loss for the FPR, which makes the training process easier. For detailed reasons, please refer to Appendix A.10.

## A.9 Extending to Multi-Target Image Segmentation

This section details the extension of our COAT (Conditional Optimization for Adaptive Thresholding) framework to address multi-target image segmentation tasks. In such applications, the goal is often to accurately identify and delineate multiple distinct object classes within an image. Unlike

| Dataset | Model | Method | $\alpha = 0.1$ | | $\alpha = 0.2$ | |
|---|---|---|---|---|---|---|
| | | | Marginal Coverage | Coverage Gap | Marginal Coverage | Coverage Gap |
| Polyp | Deeplab v3+ | CRC | 0.903 (0.008) | 0.072 (0.004) | 0.795 (0.011) | 0.133 (0.005) |
| | | COAT | 0.901 (0.004) | **0.048 (0.005)** | 0.797 (0.012) | **0.081 (0.012)** |
| | UNet | CRC | 0.899 (0.006) | 0.055 (0.002) | 0.804 (0.009) | 0.082 (0.003) |
| | | COAT | 0.901 (0.006) | **0.051 (0.006)** | 0.800 (0.011) | **0.101 (0.013)** |
| | PSPNet | CRC | 0.902 (0.010) | 0.065 (0.004) | 0.797 (0.009) | 0.107 (0.003) |
| | | COAT | 0.899 (0.005) | **0.049 (0.004)** | 0.800 (0.010) | **0.091 (0.008)** |
| | SINet | CRC | 0.899 (0.006) | 0.044 (0.002) | 0.797 (0.011) | 0.076 (0.002) |
| | | COAT | 0.900 (0.006) | **0.039 (0.002)** | 0.803 (0.008) | **0.069 (0.006)** |
| Fire | Deeplab v3+ | CRC | 0.900 (0.004) | 0.076 (0.002) | 0.801 (0.006) | 0.133 (0.002) |
| | | COAT | 0.900 (0.001) | **0.030 (0.003)** | 0.801 (0.002) | **0.047 (0.005)** |
| | UNet | CRC | 0.900 (0.003) | 0.073 (0.001) | 0.799 (0.007) | 0.127 (0.001) |
| | | COAT | 0.900 (0.001) | **0.028 (0.002)** | 0.801 (0.001) | **0.033 (0.003)** |
| | PSPNet | CRC | 0.900 (0.002) | 0.062 (0.001) | 0.800 (0.003) | 0.087 (0.001) |
| | | COAT | 0.900 (0.001) | **0.026 (0.002)** | 0.800 (0.001) | **0.033 (0.003)** |
| | SINet | CRC | 0.900 (0.001) | 0.030 (0.000) | 0.800 (0.002) | 0.038 (0.000) |
| | | COAT | 0.900 (0.001) | **0.020 (0.001)** | 0.801 (0.002) | **0.032 (0.002)** |
| Skin | Deeplab v3+ | CRC | 0.899 (0.004) | 0.091 (0.002) | 0.799 (0.007) | 0.176 (0.004) |
| | | COAT | 0.901 (0.002) | **0.054 (0.003)** | 0.800 (0.003) | **0.082 (0.003)** |
| | UNet | CRC | 0.900 (0.005) | 0.106 (0.002) | 0.801 (0.007) | 0.174 (0.003) |
| | | COAT | 0.900 (0.002) | **0.051 (0.002)** | 0.801 (0.003) | **0.073 (0.003)** |
| | PSPNet | CRC | 0.899 (0.003) | 0.081 (0.002) | 0.800 (0.007) | 0.147 (0.002) |
| | | COAT | 0.900 (0.002) | **0.040 (0.001)** | 0.801 (0.002) | **0.055 (0.003)** |
| | SINet | CRC | 0.899 (0.004) | 0.087 (0.002) | 0.801 (0.006) | 0.151 (0.002) |
| | | COAT | 0.900 (0.003) | **0.060 (0.002)** | 0.799 (0.004) | **0.081 (0.005)** |

Table 4: Marginal Coverage and Coverage Gap Results for FPR Control at $\alpha = 0.1$ and $\alpha = 0.2$ Across Different Models and Conformal Methods. Each dataset result is the mean and standard deviation of 20 random splits.

| Dataset | FNR loss weight | FPR loss weight | $\alpha = 0.1$ | | | |
|---|---|---|---|---|---|---|
| | | | Marginal Coverage (1-FNR) | Coverage Gap (1-FNR) | Marginal Coverage (1-FPR) | Coverage Gap (1-FPR) |
| Polyp | 1 | 0 | 0.896 (0.016) | 0.102 (0.010) | 0.696 (0.091) | 0.315 (0.072) |
| | 1 | 0.02 | 0.894 (0.012) | 0.103 (0.013) | 0.819 (0.047) | 0.213 (0.040) |
| | 1 | 0.2 | 0.897 (0.018) | 0.107 (0.016) | 0.815 (0.067) | 0.211 (0.054) |
| | 1 | 1 | 0.902 (0.016) | 0.140 (0.012) | 0.912 (0.016) | 0.103 (0.014) |
| | 1 | 2 | 0.897 (0.022) | 0.148 (0.016) | **0.913 (0.024)** | **0.092 (0.012)** |
| Fire | 1 | 0 | 0.900 (0.002) | 0.062 (0.001) | 0.987 (0.018) | 0.088 (0.000) |
| | 1 | 0.02 | 0.901 (0.003) | **0.059 (0.001)** | **0.988 (0.000)** | 0.088 (0.000) |
| | 1 | 0.2 | 0.900 (0.003) | 0.060 (0.001) | **0.988 (0.000)** | 0.088 (0.000) |
| | 1 | 1 | 0.899 (0.003) | 0.064 (0.006) | 0.987 (0.001) | 0.087 (0.001) |
| | 1 | 2 | 0.898 (0.005) | 0.119 (0.004) | 0.984 (0.001) | **0.084 (0.001)** |
| Skin | 1 | 0 | 0.899 (0.003) | **0.050 (0.002)** | 0.983 (0.002) | 0.094 (0.001) |
| | 1 | 0.02 | 0.900 (0.003) | 0.051 (0.001) | 0.983 (0.001) | 0.094 (0.001) |
| | 1 | 0.2 | 0.900 (0.003) | 0.051 (0.001) | 0.983 (0.001) | 0.093 (0.001) |
| | 1 | 1 | 0.901 (0.002) | 0.057 (0.012) | 0.983 (0.002) | **0.090 (0.001)** |
| | 1 | 2 | 0.899 (0.006) | 0.109 (0.003) | **0.990 (0.001)** | **0.090 (0.001)** |

Table 5: Marginal Coverage and Coverage Gap Results for FNR and FPR weighted Control at $\alpha = 0.1$. Each dataset result is the mean and standard deviation of 20 random splits.

single-class segmentation, multi-target scenarios introduce additional complexities, especially when aiming for reliable uncertainty quantification across different categories simultaneously.

### A.9.1 PROBLEM FORMULATION FOR MULTI-TARGET SEGMENTATION

For multi-target segmentation, we are interested in controlling the False Negative Rate (FNR) for a union of specified target classes. Given an input image $X_i$ and its ground truth mask $Y_i$, which now contains multiple class labels, we define a set of target class indices $\mathcal{K}_{\text{target}}$. Our objective is to construct a prediction set $\widehat{C}(X_i)$ that covers any pixel belonging to any of these target classes with a user-specified probability $1 - \alpha$. That is, we aim to control the FNR for the combined ground truth

| Dataset | FNR loss weight | FPR loss weight | $\alpha = 0.2$ | | | |
| --- | --- | --- | --- | --- | --- | --- |
| | | | Marginal Coverage (1-FNR) | Coverage Gap (1-FNR) | Marginal Coverage (1-FPR) | Coverage Gap (1-FPR) |
| Polyp | 1 | 0 | 0.796 (0.021) | **0.144 (0.013)** | 0.882 (0.034) | 0.242 (0.019) |
| | 1 | 0.02 | 0.797 (0.018) | 0.146 (0.013) | 0.895 (0.024) | 0.238 (0.015) |
| | 1 | 0.2 | 0.799 (0.014) | 0.147 (0.013) | 0.914 (0.020) | 0.224 (0.013) |
| | 1 | 1 | 0.801 (0.028) | 0.240 (0.009) | 0.982 (0.006) | 0.183 (0.004) |
| | 1 | 2 | 0.807 (0.029) | 0.242 (0.007) | **0.982 (0.004)** | **0.182 (0.004)** |
| Fire | 1 | 0 | 0.799 (0.004) | 0.080 (0.003) | 0.993 (0.000) | 0.193 (0.000) |
| | 1 | 0.02 | 0.798 (0.003) | **0.077 (0.002)** | 0.993 (0.000) | 0.193 (0.000) |
| | 1 | 0.2 | 0.800 (0.004) | **0.077 (0.002)** | 0.993 (0.000) | 0.193 (0.000) |
| | 1 | 1 | 0.799 (0.004) | 0.080 (0.003) | 0.993 (0.000) | **0.192 (0.000)** |
| | 1 | 2 | 0.800 (0.008) | 0.185 (0.008) | **0.994 (0.000)** | 0.194 (0.001) |
| Skin | 1 | 0 | 0.799 (0.004) | 0.064 (0.002) | 0.994 (0.001) | 0.195 (0.001) |
| | 1 | 0.02 | 0.801 (0.003) | 0.065 (0.001) | 0.994 (0.001) | 0.195 (0.000) |
| | 1 | 0.2 | 0.800 (0.002) | **0.064 (0.001)** | 0.994 (0.001) | 0.195 (0.000) |
| | 1 | 1 | 0.800 (0.003) | 0.067 (0.012) | 0.989 (0.002) | **0.192 (0.001)** |
| | 1 | 2 | 0.800 (0.007) | 0.167 (0.003) | **0.997 (0.000)** | 0.197 (0.000) |

Table 6: Marginal Coverage and Coverage Gap Results for FNR and FPR weighted Control at $\alpha = 0.2$. Each dataset result is the mean and standard deviation of 20 random splits.

mask $Y_i^{\text{union}} = \bigcup_{k \in \mathcal{K}_{\text{target}}} \{j : j \in Y_i \text{ and } Y_i[j] = k\}$. The FNR is then calculated as:

$$\text{FNR}(X_i) = 1 - \frac{\sum_{j=1}^{N} \widehat{C}(X_i)[j] \cdot Y_i^{\text{union}}[j]}{\sum_{j=1}^{N} Y_i^{\text{union}}[j]}$$

And we seek to achieve $\mathbb{E}[\text{FNR}(X_i)] \leq \alpha$ (marginal coverage) while minimizing the conditional coverage gap, i.e., making $\text{FNR}(X_i)$ as close to $\alpha$ as possible for individual images $X_i$.

### A.9.2 MULTI-CLASS COAT STRATEGY: PER-CLASS ADAPTIVE THRESHOLDING

To extend COAT to multi-target segmentation, we adopt a "per-class adaptive thresholding" strategy. Instead of training a single COAT model to predict a global threshold for the union of classes, we train a separate COAT model for each individual target class $k \in \mathcal{K}_{\text{target}}$. Each class-specific COAT model $f_{D,k}$ is trained to predict an optimal image-adaptive threshold $\widehat{\tau}_k(X)$ for its corresponding class's probability map $\widehat{p}_k(X)$.

The process is as follows:

- **Base Model Predictions:** A pre-trained base segmentation model (e.g., PSPNet) outputs a multi-channel probability map $\widehat{p}(X) = (\widehat{p}_1(X), \ldots, \widehat{p}_C(X))$, where $\widehat{p}_k(X)$ is the pixel-wise probability of class $k$.

- **Per-Class COAT Training:** For each target class $k \in \mathcal{K}_{\text{target}}$:
  - We extract the probability map $\widehat{p}_k(X)$ for class $k$ and the binary ground truth mask $Y_k(X)$ (where $Y_k(X)[j] = 1$ if pixel $j$ belongs to class $k$, and 0 otherwise).
  - A dedicated COAT model $f_{D,k}$ is trained using the differentiable miscoverage loss, optimizing $\widehat{\tau}_k(X)$ to achieve $1 - \alpha$ coverage for class $k$.

- **Per-Class Calibration:** Each trained $f_{D,k}$ is then calibrated using a class-specific calibration set to find a correction term $t'_k$.

- **Combined Prediction Set:** For a new test image $X$, the final prediction set $\widehat{C}(X)$ for the union of target classes is formed by taking the logical OR (union) of the individual, calibrated prediction masks from each target class:

$$\widehat{C}(X) = \bigcup_{k \in \mathcal{K}_{\text{target}}} \{j : \widehat{p}_k(X)[j] \geq \text{clip}(\widehat{\tau}_k(X) - t'_k, 0, 1)\}$$

### A.9.3 COMPARISON WITH MULTI-CLASS CRC

Our multi-class COAT method is compared with a conformal prediction baseline method. To ensure a fair comparison with our class-wise COAT strategy, we adapt this baseline method to compute an independent threshold for each target class. Specifically, this class-wise conformal prediction method computes a unique threshold $\lambda_k$ for each target class $k \in \mathcal{K}_{\text{target}}$ by calibrating specifically

for that class's False Negative Rate (FNR) on the calibration set. Subsequently, these class-specific thresholds $\lambda_k$ are applied to the probability maps of their respective target classes.

The final prediction set for this baseline method is formed by combining the predictions of individual classes:

$$\widehat{C}_{\text{CRC}}(X) = \bigcup_{k \in \mathcal{K}_{\text{target}}} \{j : \widehat{p}_k(X)[j] \geq \lambda_k\}$$

It is important to note that while this class-wise thresholding approach can provide *class-marginal coverage* guarantees (i.e., independently controlling each class's FNR on the calibration set), it does not inherently guarantee *image-conditional coverage* or *class-conditional coverage*.

Therefore, the key distinction lies in the adaptability and scope of guarantees. While this class-wise baseline method provides marginal guarantees for each class's FNR, its reliance on static, class-specific thresholds inherently limits its ability to control image- and class-specific conditional coverage. Our class-wise COAT strategy, by training an adaptive threshold predictor for each class, aims to provide more fine-grained, image-conditional FNR control for each target class. When these individually optimized class predictions are combined, we expect to achieve tighter control over the class-union conditional FNR, thereby closing the coverage gap.

A.9.4    MULTI-CLASS IMAGE SEGMENTATION RESULTS: CRC VS. COAT

We evaluate our multi-class COAT on the Cityscapes dataset (Cordts et al., 2016), a challenging benchmark for urban scene understanding. We selected three diverse target classes:

- **Road (index 0):** A common and extensive background class.
- **Person (index 11):** A critical and often smaller foreground class.
- **Car (index 13):** Another important foreground class, varying in size and shape.

Our experiments are conducted with a target FNR level of $\alpha = 0.1$. To ensure robustness, we perform 20 random splits of the dataset for training, calibration, and testing, and report mean and standard deviation for performance metrics. The base segmentation model used is PSPNet, pre-trained on Cityscapes.

| Metric | CRC | COAT |
|---|---|---|
| Coverage | 0.9152 (0.0091) | 0.9048 (0.0107) |
| Coverage Gap | 0.0844 (0.0048) | **0.0811 (0.0150)** |
| Coverage (road) | 0.9031 (0.0113) | 0.8978 (0.0127) |
| Coverage gap (road) | 0.0978 (0.0068) | **0.0938 (0.0188)** |
| Coverage (person) | 0.9044 (0.0139) | 0.9017 (0.0155) |
| Coverage gap (person) | 0.1069 (0.0075) | **0.0997 (0.0088)** |
| Coverage (car) | 0.9035 (0.0092) | 0.9013 (0.0113) |
| Coverage gap (car) | 0.0827 (0.0029) | **0.0670 (0.0036)** |

Table 7: Performance Comparison of CRC and COAT on Cityscapes Multi-Target Segmentation

As shown in Table 7, compared with CRC, COAT not only achieves smaller conditional coverage and more stable marginal coverage across categories, but also attains smaller category-conditional coverage.

**Remark.** *As shown in Figure 11, our multi-class COAT strategy demonstrates promising results in extending adaptive thresholding to multi-target image segmentation, achieving better class-conditional coverage compared to the baseline. However, it is important to acknowledge certain considerations for its broader application. The current approach involves training a separate COAT model for each target class. While effective, this can become computationally intensive and time-consuming when dealing with a very large number of target classes. Future work will explore the design of a joint loss function that enables simultaneous adaptive thresholding for multiple classes within a single optimization framework, aiming to improve efficiency without compromising performance. Additionally, the effectiveness of COAT, like other conformal prediction methods, relies on*

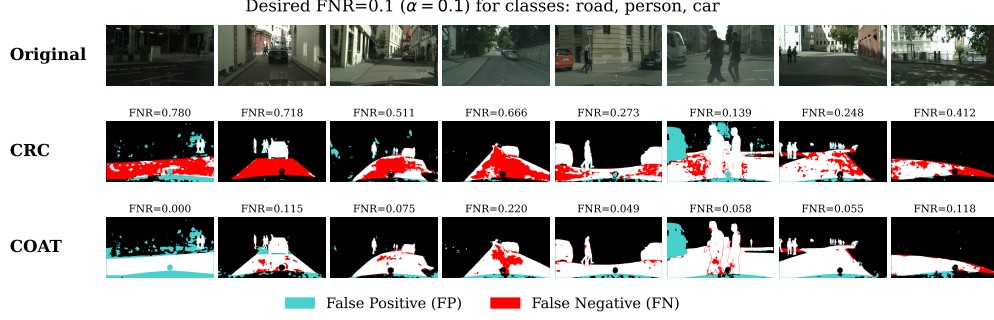

Figure 11: Qualitative comparison of CRC and COAT prediction sets at a significance level of $\alpha = 0.1$.

*the assumption of exchangeability. In scenarios with extremely limited datasets, this assumption may be compromised, potentially leading to a degradation in the method's performance and reliability.*

### A.10 DUAL COAT FOR SYNCHRONOUS FNR AND FPR CONTROL

Simultaneously controlling False Negative Rate (FNR) and False Positive Rate (FPR) is crucial in safety-critical applications. While our COAT framework primarily targets FNR by minimizing coverage gaps, its differentiable nature allows direct extension to FPR control, enabling a flexible approach to balance both risks with a single threshold.

#### A.10.1 PROBLEM FORMULATION FOR DUAL CONTROL

In image segmentation, a predicted set $\widehat{C}(X, \tau)$ is defined by a base model's probability map $\widehat{p}(X)$ and a threshold $\tau$:

$$\widehat{C}(X, \tau) = \{j : \widehat{p}_j(X) \geq \tau\}$$

- **False Negative Rate (FNR):** Proportion of true positive pixels incorrectly excluded.

$$\text{FNR}(X, \tau) = 1 - \frac{\sum_{j=1}^{N} \widehat{C}(X, \tau)[j] \cdot Y[j]}{\sum_{j=1}^{N} Y[j]}$$

  Objective: $\text{FNR}(X, \tau) \leq \alpha_{FNR}$. Lower $\tau$ reduces FNR. FNR is non-increasing with $\tau$.

- **False Positive Rate (FPR):** Proportion of true negative pixels incorrectly included. For the binary mask representation where $Y \in \{0, 1\}^N$, the complement is $1 - Y[j]$ for each pixel $j$.

$$\text{FPR}(X, \tau) = \frac{\sum_{j=1}^{N} \widehat{C}(X, \tau)[j] \cdot (1 - Y[j])}{\sum_{j=1}^{N} (1 - Y[j])}$$

  Objective: $\text{FPR}(X, \tau) \leq \alpha_{FPR}$. Higher $\tau$ reduces FPR. FPR is non-decreasing with $\tau$.

The challenge lies in finding a single $\tau$ that satisfies both $\text{FNR}(X, \tau) \leq \alpha_{FNR}$ and $\text{FPR}(X, \tau) \leq \alpha_{FPR}$, due to their opposing dependencies on $\tau$.

#### A.10.2 INDIVIDUAL COAT MODELS FOR FNR AND FPR

We train two independent COAT models, each optimized for a specific risk by learning image-adaptive thresholds:

1. **COAT$_{\text{FNR}}$:** Learns $\widehat{\tau}_{FNR}(X)$ aiming to achieve $\text{FNR}(X, \widehat{\tau}_{FNR}(X)) \approx \alpha_{FNR}$.
2. **COAT$_{\text{FPR}}$:** Learns $\widehat{\tau}_{FPR}(X)$ aiming to achieve $\text{FPR}(X, \widehat{\tau}_{FPR}(X)) \approx \alpha_{FPR}$.

### A.10.3   INTERPOLATION FOR JOINT MARGINAL RISK CONTROL

To balance overall (marginal) FNR and FPR across the dataset, we propose an interpolation strategy:

1. **Obtain Image-Adaptive Thresholds:** Train COAT$_{\text{FNR}}$ and COAT$_{\text{FPR}}$ independently, then obtain calibrated thresholds $\tau'_{FNR}(X)$ and $\tau'_{FPR}(X)$ for all test images using the calibration procedure from Algorithm 1.

2. **Linear Interpolation:** For each image $X$, define an interpolated threshold $\tau_\lambda(X)$ using $\lambda \in [0, 1]$:
$$\tau_\lambda(X) = \lambda \cdot \tau'_{FPR}(X) + (1 - \lambda) \cdot \tau'_{FNR}(X)$$
This transitions between FNR-priority ($\lambda = 0$) and FPR-priority ($\lambda = 1$) thresholds.

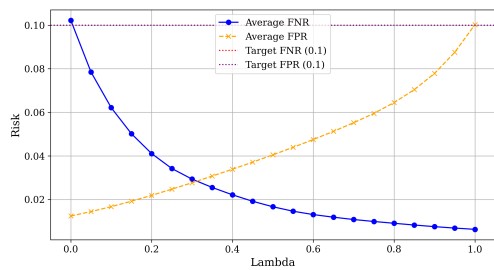 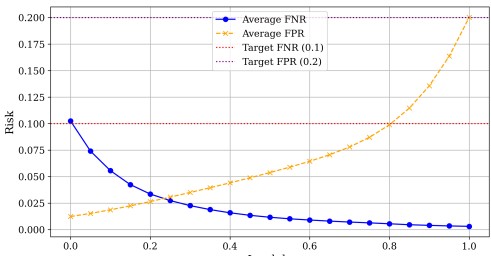

Figure 12: Left: $\alpha_{FNR} = 0.1$, $\alpha_{FPR} = 0.1$; Right: $\alpha_{FNR} = 0.1$, $\alpha_{FPR} = 0.2$. On the fire dataset, when $\lambda$ takes any value within the interval $[0, 1]$, we can simultaneously control both the FNR and FPR.

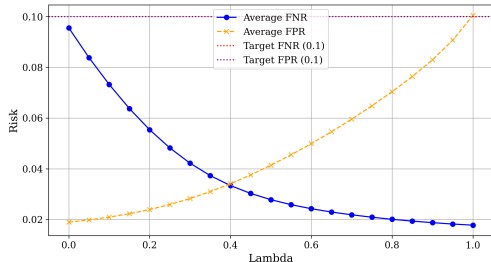 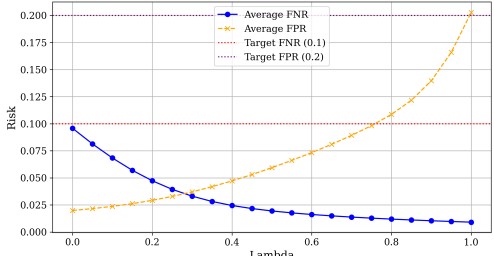

Figure 13: Left: $\alpha_{FNR} = 0.1$, $\alpha_{FPR} = 0.1$; Right: $\alpha_{FNR} = 0.1$, $\alpha_{FPR} = 0.2$. On the skin dataset, when $\lambda$ takes any value within the interval $[0, 1]$, we can simultaneously control both the FNR and FPR.

3. **Marginal Risk Curve Generation:** Iterate $\lambda$ from 0 to 1. For each $\lambda$, compute $\tau_\lambda(X)$ for all test images. Then, for each image $X$, construct the prediction set $\widehat{C}(X, \tau_\lambda(X)) = \{j : \widehat{p}_j(X) \geq \tau_\lambda(X)\}$. Finally, calculate the average (marginal) FNR and FPR across the test set using these prediction sets. Plot these average FNR and FPR values.

4. **Interpretation and User Guidance:** The resulting curve visualizes the FNR-FPR trade-off:

   - $\lambda = 0$: $\tau_\lambda(X) = \tau'_{FNR}(X)$. Expected marginal FNR near $\alpha_{FNR}$, FPR potentially higher.
   - $\lambda = 1$: $\tau_\lambda(X) = \tau'_{FPR}(X)$. Expected marginal FPR near $\alpha_{FPR}$, FNR potentially higher.
   - As $\lambda$ increases, $\tau_\lambda(X)$ generally increases, causing marginal FNR to typically increase and marginal FPR to typically decrease.

   This curve allows users to select an optimal $\lambda$ to balance marginal FNR and FPR according to application-specific safety and performance needs, aiming for a region where both risks are at or below their target $\alpha$ levels.

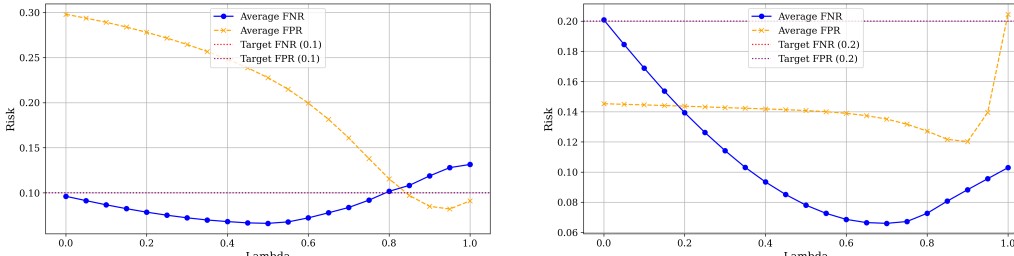

Figure 14: Left: $\alpha_{FNR} = 0.1$, $\alpha_{FPR} = 0.1$; Right: $\alpha_{FNR} = 0.2$, $\alpha_{FPR} = 0.2$. On the polyp dataset, due to its greater difficulty and the extremely small dataset sample size, the basic model yields poor prediction performance. This makes it impossible for us to control both the FNR and FPR simultaneously. As we increase our $\alpha$, as shown in the right panel, we become able to control both FNR and FPR concurrently.

### A.10.4 EXPERIMENTAL ANALYSIS

**Remark.** *Note that Figures 12, 13, and 14, which plot average FNR/FPR versus λ, are intended to serve as **empirical guidance** for applying this method to a new sample X. For any such sample, our method computes two calibrated thresholds: $\tau'_{FNR}$, designed to satisfy $\mathbb{E}(FNR) \leq \alpha_{FNR}$, and $\tau'_{FPR}$, designed to satisfy $\mathbb{E}(FPR) \leq \alpha_{FPR}$. This leads to two scenarios:*

- ***Feasible Solution** ($\tau'_{FPR} \leq \tau'_{FNR}$): A feasible interval exits. Any threshold $\tau$ selected from this interval **guarantees** that both risk constraints ($\mathbb{E}(FNR) \leq \alpha_{FNR}$ and $\mathbb{E}(FPR) \leq \alpha_{FPR}$) are satisfied. In this case, $\lambda$ is a preference parameter to choose a point within this guaranteed-safe region, guided by the average performance trends shown in the plots.*

**Desired FNR=FPR=0.2 (alpha =0.2)**

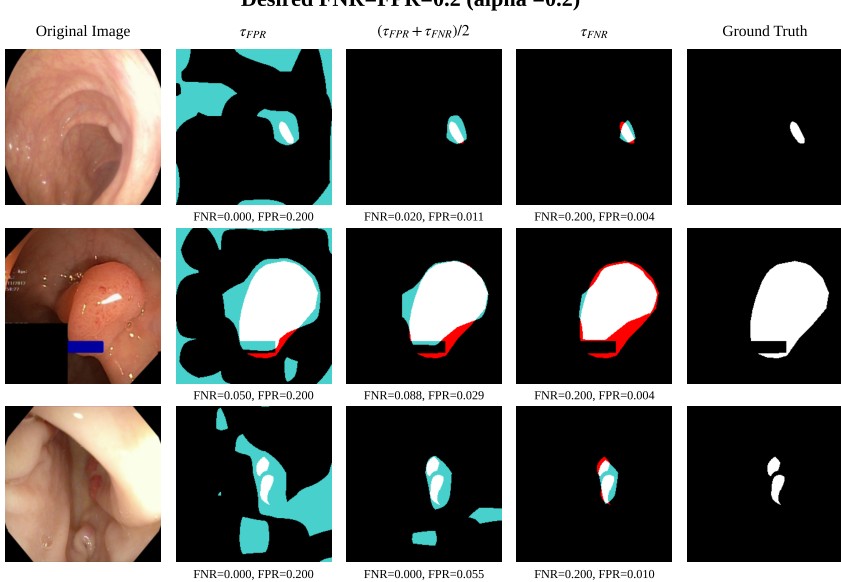

Figure 15: Visualization of results for different $\lambda$ values on the polyp dataset when $\alpha_{FNR} = \alpha_{FPR} = 0.2$.

- ***No Feasible Solution** ($\tau'_{FPR} > \tau'_{FNR}$): No single threshold can satisfy both constraints, necessitating a trade-off. Here, $\lambda$ directly governs this choice: (1) $\lambda = 0$ selects $\tau'_{FNR}$, guaranteeing average FNR control. The plot provides an empirical estimate of the resulting (violated) FPR; and (2) $\lambda = 1$ selects $\tau'_{FPR}$, guaranteeing average FPR control. The plot estimates the resulting FNR. Values of $\lambda$ between 0 and 1 represent a smooth interpolation*

*between these two strategies. However, any choice of $\lambda \in (0, 1)$ **does not guarantee** control over either the FNR or the FPR in this case.*

*In addition, two key points deserve emphasis. First, the risk control discussed here refers to **marginal (average) risk control** over the data distribution, not a conditional guarantee for individual samples. Second, the curves show the expected trend; the actual FNR/FPR achieved by applying a $\lambda$-chosen threshold may not lie exactly on the curve. Consequently, while this method provides a flexible decision-support tool, the above considerations should be carefully weighed, particularly in the infeasible regime. When simultaneous risk control is possible, however, it does enable empirical performance optimization within a guaranteed safe region.*

When $\lambda$ equals 0 in Figures 12, 13, and 14, the corresponding scenarios are equivalent to the results in Tables 5 and 6 where the ratio of FNR loss to FPR loss is $1 : 0$. In the polyp dataset (Figure 14), there exists no value of $\lambda$ that can simultaneously ensure the marginal coverage of both FNR and FPR. When we employ the weighted loss, it significantly enhances the performance of the COAT model. Conversely, in Figures 12 and 13, where FNR and FPR are guaranteed to be less than or equal to $\alpha$, there is no substantial improvement in performance with the weighted loss.

**Desired FNR=FPR=0.2 ($\alpha$=0.2)**

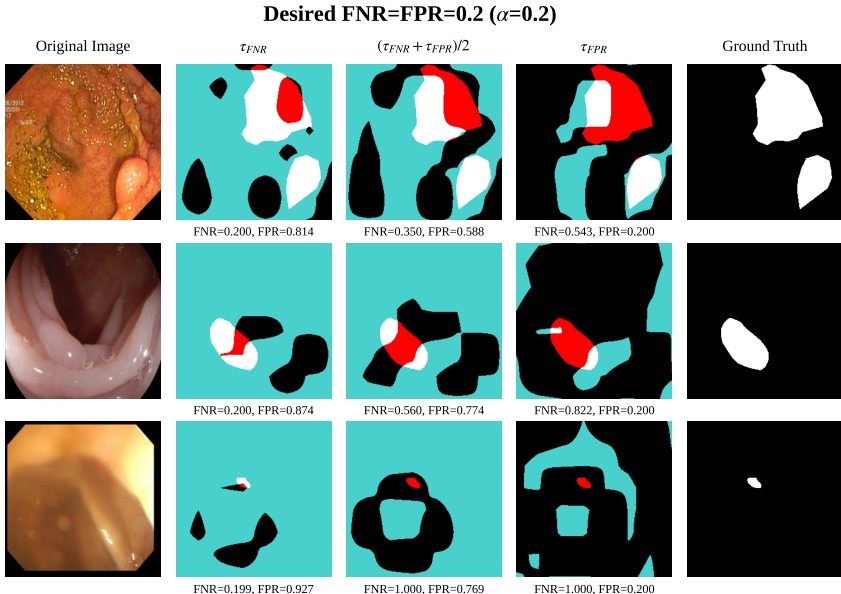

Figure 16: Visualization of results for different $\lambda$ values on the polyp dataset when $\alpha_{FNR} = \alpha_{FPR} = 0.2$.

As shown in Figure 15, we have visualized the results for different values of lambda on the polyp dataset when $\alpha_{FNR} = \alpha_{FPR} = 0.2$. The 2nd to 4th columns represent the results for $\lambda = 1, 0.5,$ and 0, respectively. All of them simultaneously control both the FNR and FPR below 0.2. Users can select different values of lambda within the range based on their preference for other metrics. As shown in Figure 16, it presents an opposite scenario on the polyp dataset, that is, for certain images, there is no feasible solution, and we have to make a choice between controlling the FNR or the FPR.

## A.11 COAT CORE CODE IMPLEMENTATION

To enhance the reproducibility of our work, this appendix provides the core PyTorch implementation details for our COAT (Conditional Optimization for Adaptive Thresholding) method. We have split the implementation into two parts, mirroring the two main stages of the framework: the end-to-end training of the threshold predictor, and the subsequent post-hoc calibration and inference process.

### A.11.1 ALGORITHM FOR COAT TRAINING PHASE

This first algorithm details the training procedure for the adaptive threshold predictor, $f_D$. The core components are the network architecture itself and the novel differentiable miscoverage loss function, $L_{\text{COAT}}$, which enables direct, gradient-based optimization towards the conditional coverage target.

---

**Algorithm 2 Python Pseudo-Code for COAT: Training Phase.** This code outlines the end-to-end training of the `ThresholdPredictor` network. It includes the network architecture, the differentiable TPR approximation (`calculate_differentiable_tpr`), and the main training loss computation within a single training step function (`coat_training_step`).

---

```python
import torch
import torch.nn as nn
from torchvision.models import resnet50, ResNet50_Weights

class ThresholdPredictor(nn.Module):
    """The adaptive threshold predictor network, f_D in the paper."""
    def __init__(self, pretrained: bool = True):
        super().__init__()
        self.resnet = resnet50(weights=ResNet50_Weights.DEFAULT)
        # Modify input layer for 4 channels (Image RGB + Probability Map)
        original_conv1 = self.resnet.conv1
        self.resnet.conv1 = nn.Conv2d(4, 64, kernel_size=7, stride=2, padding=3, bias=False)
        with torch.no_grad():
            self.resnet.conv1.weight[:, :3, :, :] = original_conv1.weight.clone()
            self.resnet.conv1.weight[:, 3, :, :] = original_conv1.weight.clone().mean(dim=1,
            keepdim=True)
        # Modify output layer for a single threshold value
        self.resnet.fc = nn.Linear(self.resnet.fc.in_features, 1)

    def forward(self, image: torch.Tensor, prob_map: torch.Tensor) -> torch.Tensor:
        input_tensor = torch.cat([image, prob_map.unsqueeze(1)], dim=1)
        return torch.sigmoid(self.resnet(input_tensor))

def calculate_differentiable_tpr(
    prob_map: torch.Tensor, true_mask: torch.Tensor, pred_tau: torch.Tensor,
    T: float, epsilon: float) -> torch.Tensor:
    """Computes the differentiable TPR'."""
    soft_mask = torch.sigmoid((prob_map - pred_tau) / T)
    intersection = torch.sum(soft_mask * true_mask)
    true_mask_size = torch.sum(true_mask)
    return intersection / (true_mask_size + epsilon)

def coat_training_step(
    model: ThresholdPredictor, optimizer: torch.optim.Optimizer,
    image_batch: torch.Tensor, prob_map_batch: torch.Tensor, true_mask_batch: torch.Tensor,
    alpha: float, T: float, epsilon: float):
    """Performs a single training step for the COAT model."""
    model.train()
    optimizer.zero_grad()

    # Predict image-specific thresholds
    pred_taus = model(image_batch, prob_map_batch)

    total_loss = 0.0
    target_coverage = 1 - alpha
    # Compute loss for each sample in the batch
    for i in range(image_batch.size(0)):
        tpr_prime = calculate_differentiable_tpr(
            prob_map_batch[i], true_mask_batch[i], pred_taus[i], T, epsilon)
        # L_COAT = (TPR' - (1 - alpha))^2
        loss = (tpr_prime - target_coverage)**2
        total_loss += loss

    # Update parameters
    avg_loss = total_loss / image_batch.size(0)
    avg_loss.backward()
    optimizer.step()
    return avg_loss.item()
```

---

### A.11.2 ALGORITHM FOR COAT CALIBRATION AND INFERENCE PHASE

Once the 'ThresholdPredictor' is trained, this second algorithm details the procedure for calibration and inference. A held-out calibration set, $\mathcal{D}_{cal}$, is used to compute a single, global correction term $t'$. This correction is then applied to the predicted thresholds for all test images to generate the final prediction sets, $C(X)$, which are guaranteed to satisfy the marginal coverage property.

**Algorithm 3 Python Pseudo-Code for COAT: Calibration and Inference Phase.** This code shows the post-training procedure. First, `calibrate_coat` uses the trained model and a calibration set to find the optimal correction term `t_prime`. Then, `apply_calibrated_coat` uses this term to generate final prediction sets on the test data.

```python
import torch
import numpy as np

def calibrate_coat(
    model: torch.nn.Module, cal_loader: torch.utils.data.DataLoader, alpha: float) -> float:
    """Performs calibration to find the correction term t'."""
    model.eval()
    base_taus_cal = []

    # Compute base thresholds for the calibration set D_cal
    with torch.no_grad():
        for (images, prob_maps, _) in cal_loader:
            base_taus_cal.extend(model(images, prob_maps).cpu().numpy())

    # Define the empirical coverage function R(t)
    def get_empirical_coverage(t: float) -> float:
        coverages = []
        cal_dataset = cal_loader.dataset
        for i in range(len(cal_dataset)):
            _, prob_map, true_mask = cal_dataset[i]
            # Apply correction: tau_i - t
            adjusted_tau = np.clip(base_taus_cal[i] - t, 0, 1)
            pred_set = prob_map >= adjusted_tau
            if true_mask.sum() > 0:
                coverage = (pred_set & true_mask).sum() / true_mask.sum()
                coverages.append(coverage.item())
        return np.mean(coverages) if coverages else 0.0

    # Find minimal correction t' via search (e.g., binary search)
    # Target is R(t) >= (|D_cal|+1)/|D_cal| * (1-alpha)
    n = len(cal_loader.dataset)
    target_cal_coverage = (n + 1) / n * (1 - alpha)
    # A placeholder for a binary search function to find t'
    t_prime = binary_search_for_t(get_empirical_coverage, target_cal_coverage)

    return t_prime

def apply_calibrated_coat(
    model: torch.nn.Module, test_loader: torch.utils.data.DataLoader, t_prime: float) -> list:
    """Applies the calibrated model to the test set."""
    model.eval()
    prediction_sets = []
    with torch.no_grad():
        for (images, prob_maps, _) in test_loader:
            # Compute base thresholds for test images
            base_taus_test = model(images, prob_maps)
            for i in range(len(images)):
                # Calculate the adjusted threshold
                adjusted_tau = torch.clamp(base_taus_test[i] - t_prime, 0, 1)
                # Generate the final prediction set
                pred_set = prob_maps[i] >= adjusted_tau
                prediction_sets.append(pred_set)

    return prediction_sets # Line 23: Output C(X_i)
```

