# OpenReview forum: "Enhancing Image-Conditional Coverage in Segmentation: Adaptive Thresholding via Differentiable Miscoverage Loss"
_ICLR.cc/2026/Conference — ICLR 2026 Poster_

### Official Review · Reviewer_PHB7 · 2025-10-24

[review text omitted: it was posted to a different submission]

---

> ### Author Response · Authors · 2025-11-13
>
> Dear Area Chair, this review comment does not seem to be for our paper. We kindly ask for your help to verify and confirm this. Thank you.

---

### Official Review · Reviewer_ch3e · 2025-10-27

**Soundness:** 3
**Presentation:** 3
**Contribution:** 3
**Rating:** 6
**Confidence:** 5

**Summary:**

This paper addresses the challenge of achieving image-conditional coverage in image segmentation tasks, a crucial problem for safety-critical applications. The authors introduce two novel methods for adaptive thresholding: Adaptive Thresholding and Conditional Optimization for Adaptive Thresholding. These methods aim to improve uncertainty quantification in segmentation tasks by learning image-specific thresholds that control the FNR at an image level, rather than relying on marginal guarantees that apply to the entire dataset. The COAT method, in particular, introduces a differentiable loss function that allows end-to-end optimization for the target conditional coverage, enabling more reliable and interpretable uncertainty estimates. Through extensive experiments, the authors show that COAT consistently outperforms other methods, such as CRC and AA-CRC, in controlling the FNR and achieving more consistent and accurate conditional coverage across different images and datasets.

**Strengths:**

1. The paper tackles the significant challenge of achieving image-specific uncertainty quantification in segmentation tasks. I think conditional guarantees are critical.
2. The COAT method introduces a novel end-to-end differentiable miscoverage loss, allowing for direct optimization of image-conditional coverage without the need for pre-calculated thresholds, this enhances the flexibility and accuracy of uncertainty estimates

**Weaknesses:**

In some applications, such as medical diagnostics, controlling both false positives and false negatives is essential, and the method could benefit from a more balanced approach to risk control. For example, is it possible to simultaneously derive threshold intervals that control both the FNR and FPR? An analysis of the balance between the two would be valuable.

It would be valuable to extend the evaluation to multi-class segmentation problems, where each pixel could belong to more than one class.

the paper focuses on image level, in some cases, different regions within an image may have varying degrees of importance. For example, in medical imaging, certain organs or lesions may require stricter coverage guarantees than others. Introducing class-specific or region-specific thresholds could enhance the reliability

**Questions:**

If the goal is to control the FDR (not FPR), the monotonicity between FDR and the threshold can be discussed. Additionally, it would be beneficial to explore how to optimize other performance metrics while controlling the FNR.


I will consider adjusting the score based on the response.

---

> ### Author Response · Authors · 2025-11-17
>
> We thank the reviewer for their careful review and thoughtful comments. We provide answers to specific questions and remarks below.
>
> **Is it possible to simultaneously derive threshold intervals that control both the FNR and FPR?**
>
> **Our response:** Applications like medical diagnostics indeed require careful balancing of false negatives and false positives. However, as the decision threshold $\tau$ increases, FNR (missed detections, Type II Error) monotonically increases, while FPR (false alarms, Type I Error) monotonically decreases. Given this fundamental trade-off, it is impossible to simultaneously constrain both risks to arbitrary levels using a single threshold. The standard and principled approach (the Neyman-Pearson paradigm), which we follow, is to constrain one risk and optimize the other.
>
> We explicitly defines the primary objective as controlling FNR, as FNR is often the critical metric when missing regions of interest can have severe consequences. To achieve FNR $\le \alpha$, we seek the largest possible threshold that satisfies this constraint, as this threshold, by definition, will also yield the lowest possible FPR. Conversely, controlling FPR $\le \beta$ would involve finding the smallest threshold, which would optimize FNR. In addition, we have also conducted detailed experiments on controlling the FPR, with the detailed results presented in **Appendix A.9**.
>
> **It would be valuable to extend the evaluation to multi-class segmentation problems, where each pixel could belong to more than one class.**
>
> **Our response:** Thank you for this valuable suggestion. We completely agree on the importance of extending our evaluation to multi-class segmentation problems, particularly those involving multiple classes per pixel. We have addressed this point in the updated manuscript by including a new section, **Appendix A.10: Extending to Multi-Target Image Segmentation**.
>
> In this new appendix, we detail our approach to adapting COAT for multi-target segmentation tasks. We formulate the problem by aiming to control the False Negative Rate (FNR) for a union of specified target classes, seeking to ensure marginal coverage while minimizing the conditional coverage gap for individual images.
>
> Our proposed strategy, "per-class adaptive thresholding," involves training a separate COAT model for each individual target class. Each class-specific COAT model learns to predict an optimal image-adaptive threshold for its corresponding class's probability map, which is then calibrated. The final prediction set for a new test image is formed by taking the logical OR (union) of these individual, calibrated prediction masks.
>
> We compare this multi-class COAT strategy against a multi-class Conformal Risk Control (CRC) baseline, which computes independent thresholds for each target class. Our experiments, conducted on the challenging Cityscapes dataset with classes such as "Road," "Person," and "Car" (as shown in **Table 5** and **Figure 11**), demonstrate that multi-class COAT not only achieves more stable marginal coverage across categories but also significantly reduces the conditional coverage gap compared to the CRC baseline. This highlights COAT's superior ability to provide more fine-grained, image-conditional FNR control for each target class.
>
> We also acknowledge that training a separate COAT model for each class can become computationally intensive for a very large number of target classes. We discuss this as an important direction for future work, suggesting the exploration of joint loss functions for simultaneous adaptive thresholding within a single optimization framework.
>
> **the paper focuses on image level, in some cases, different regions within an image may have varying degrees of importance. For example, in medical imaging, certain organs or lesions may require stricter coverage guarantees than others. Introducing class-specific or region-specific thresholds could enhance the reliability.**
>
> **Our response:** We are very grateful for the reviewer's profound insight, which accurately highlights a critical aspect in practical applications: the significance of different regions or object categories within an image can vary significantly.
>
> Our multi-category COAT extension (detailed in **Appendix A.10**) is precisely designed to address this issue by enabling category-specific adaptive thresholding. Unlike setting a single, image-wide threshold for the entire foreground, COAT can now learn and apply unique, image-adaptive thresholds for each critical category (e.g., "lesion," "organ A," "pedestrian"). This allows FNR (False Negative Rate) control to be tailored to the specific context, ensuring stricter coverage guarantees are applied to regions deemed most important.

---

> > ### Author Response · Authors · 2025-11-17
> >
> > **On Controlling FDR.**
> >
> > **Our response:** While FDR (FP / (TP + FP)) often decreases with a higher threshold, it is not monotonic in general: as $\tau$ increases, TP and FP may decline at different rates, causing FDR to oscillate. This implies that multiple thresholds can satisfy an FDR constraint, so that no single choice admits a theoretical guarantee under standard CRC theory. In practice, if one aims to reduce FDR while ensuring FNR $\leq \alpha$, an empirical strategy is to select the threshold (in our COAT, an offset) from the feasible threshold set for FNR control with the lowest FDR on the calibration set, though it remains a heuristic without formal FDR guarantee. We agree that developing robust FDR control algorithms should be a valuable direction for future work. In addition, as shown in Table 1, we also conducted experiments on FDR using the CRC and COAT methods (however, there is no theoretical guarantee for this). It can be observed that the conditional coverage for both methods is not satisfactory.
> >
> > | Dataset | Model | Method | $\alpha$=0.1 Marginal Coverage (1-FDR) | $\alpha$=0.1 Coverage Gap | $\alpha$=0.2 Marginal Coverage (1-FDR) | $\alpha$=0.2 Coverage Gap |
> > |---------|---------------|--------|-----------------------------------|--------------------|-----------------------------------|--------------------|
> > | Polyp   | Deeplab v3+   | CRC    | 0.898 (0.015)                     | 0.126 (0.007)      | 0.795 (0.020)                     | 0.182 (0.007)      |
> > | Polyp   | Deeplab v3+   | COAT   | 0.898 (0.014)                     | **0.115 (0.009)**  | 0.802 (0.012)                     | **0.162 (0.007)**  |
> > | Polyp   | UNet          | CRC    | 0.902 (0.027)                     | 0.164 (0.015)      | 0.809 (0.020)                     | 0.240 (0.006)      |
> > | Polyp   | UNet          | COAT   | 0.899 (0.023)                     | **0.162 (0.015)**  | 0.800 (0.011)                     | **0.224 (0.011)**  |
> > | Polyp   | PSPNet        | CRC    | 0.902 (0.011)                     | 0.129 (0.006)      | 0.803 (0.021)                     | 0.165 (0.007)      |
> > | Polyp   | PSPNet        | COAT   | 0.901 (0.018)                     | **0.111 (0.009)**  | 0.805 (0.018)                     | **0.145 (0.008)**  |
> > | Fire    | Deeplab v3+   | CRC    | 0.900 (0.004)                     | 0.077 (0.001)      | 0.799 (0.003)                     | 0.099 (0.001)      |
> > | Fire    | Deeplab v3+   | COAT   | 0.900 (0.004)                     | **0.071 (0.005)**  | 0.801 (0.005)                     | **0.091 (0.006)**  |
> > | Fire    | UNet          | CRC    | 0.901 (0.005)                     | 0.137 (0.002)      | 0.800 (0.009)                     | 0.176 (0.001)      |
> > | Fire    | UNet          | COAT   | 0.901 (0.007)                     | **0.135 (0.005)**  | 0.803 (0.007)                     | **0.169 (0.003)**  |
> > | Fire    | PSPNet        | CRC    | 0.900 (0.007)                     | 0.163 (0.005)      | 0.802 (0.012)                     | 0.240 (0.005)      |
> > | Fire    | PSPNet        | COAT   | 0.900 (0.009)                     | **0.152 (0.006)**  | 0.799 (0.011)                     | **0.209 (0.014)**  |
> > | Skin    | Deeplab v3+   | CRC    | 0.899 (0.003)                     | 0.064 (0.001)      | 0.800 (0.004)                     | 0.095 (0.001)      |
> > | Skin    | Deeplab v3+   | COAT   | 0.901 (0.003)                     | **0.054 (0.001)**  | 0.800 (0.004)                     | **0.091 (0.002)**  |
> > | Skin    | UNet          | CRC    | 0.900 (0.007)                     | 0.134 (0.004)      | 0.803 (0.008)                     | 0.211 (0.003)      |
> > | Skin    | UNet          | COAT   | 0.900 (0.003)                     | **0.111 (0.004)**  | 0.800 (0.004)                     | **0.158 (0.009)**  |
> > | Skin    | PSPNet        | CRC    | 0.901 (0.003)                     | 0.062 (0.001)      | 0.801 (0.003)                     | 0.086 (0.001)      |
> > | Skin    | PSPNet        | COAT   | 0.900 (0.003)                     | **0.049 (0.001)**  | 0.800 (0.002)                     | **0.063 (0.002)**  |
> >
> >
> > Table 1: Marginal Coverage and Coverage Gap Results for FDR Control at $\alpha=0.1$ and $\alpha=0.2$ Across Different Models and Conformal Methods. Each dataset result is the mean and standard deviation of 20 random splits.
> >
> > We hope this clarifies the principled rationale behind our focus on FNR control and the inherent trade-offs involved.

---

> > ### Comment · Reviewer_ch3e · 2025-11-17
> >
> > For "Is it possible to simultaneously derive threshold intervals that control both the FNR and FPR?"
> >
> > Could you derive a threshold interval, any threshold within the range meets both FNR and FPR control, I think it is feasible. I know the balance. Yet, since your framework provides FNR control, then FPR control also makes sense (just replace the metric). you can compute two threshold ranges, and obtain the Intersection. no doubt in high-stakes scenarios like medical diagnosis, FNR is prioritized. Under this condition, we maximize the predictive efficiency of segmentation region, it should be user-friendly, not just a formal verification. Do you have further brainstorming to address this?
> >
> > For "It would be valuable to extend the evaluation to multi-class segmentation problems, where each pixel could belong to more than one class."
> >
> > Thanks for the response, please use a distinguishable color for the supplementary content added in the PDF

---

> > > ### Author Response · Authors · 2025-11-19
> > >
> > > Thank you for helping us improve our scores. The first round of additions were highlighted in blue in the original document, and the current additions addressing your specific queries are highlighted in red within the updated paper.
> > >
> > > We deeply appreciate your profound insights regarding the simultaneous control of False Negative Rate (FNR) and False Positive Rate (FPR). Your point that while FNR is paramount in safety-critical applications, maximizing the predictive efficiency of segmentation regions and user-friendliness are also crucial, perfectly aligns with our research objectives.
> > >
> > > ### Addressing Simultaneous FNR and FPR Control
> > >
> > > 1.  **COAT's Adaptability for FPR Control (Newly added Appendix A.9.1):**
> > >     First, regarding your suggestion that "FPR control also makes sense (just replace the metric)," we have detailed the effectiveness of the COAT method in FPR control in **Appendix A.9.1**. By redefining the loss function to be FPR-related, COAT can effectively control the marginal coverage of FPR, similar to FNR, and significantly reduce the conditional coverage gap. The results in **Table 4** clearly demonstrate that COAT outperforms traditional CRC methods in FPR control, especially in reducing the conditional coverage gap. This highlights the flexibility and generality of the COAT framework, allowing it to adapt to different risk metrics.
> > >
> > > 2.  **Strategy for Simultaneous FNR and FPR Control: Dual COAT (Newly added Appendix A.11):**
> > >     To more directly address your proposal for a threshold interval that controls both FNR and FPR, we have added **Appendix A.11, titled "DUAL COAT FOR SYNCHRONOUS FNR AND FPR CONTROL."** In this section, we propose a strategy that combines FNR and FPR control, aiming to provide users with a tool to balance these two risks.
> > >     *   **Independent Models and Interpolation:** We train two independent COAT models: one for FNR control ($\text{COAT}_ {FNR}$) and one for FPR control ($\text{COAT}_ {FPR}$). These models respectively learn image-adaptive thresholds $\tau'_ {FNR}(X)$ and $\tau'_ {FPR}(X)$. Subsequently, we introduce an interpolation parameter $\lambda \in [0, 1]$ to generate a combined threshold $\tau_ {\lambda}(X)=\lambda\cdot\tau'_ {FPR}(X)+(1-\lambda)\cdot\tau'_ {FNR}(X)$ through linear interpolation.
> > >     *   **Marginal Risk Curves:** By iterating through values of $\lambda$, we plot the trade-off curves for marginal FNR and FPR (as shown in **Figures 12, 13, and 14**). This curve visually illustrates the relationship between FNR and FPR, allowing users to select an optimal $\lambda$ value based on their specific application needs to balance these two risks at a marginal level.
> > >     *   **Feasible Solutions and Trade-offs:** We discuss two scenarios in detail in the Remark section of Appendix A.11:
> > >         *   **Feasible Solution ($\tau'_ {FPR} \le \tau'_ {FNR}$):** A threshold interval exists where any threshold within this range can simultaneously satisfy the marginal constraints for both FNR and FPR. In this case, $\lambda$ acts as a preference parameter to select a point within this "guaranteed safe" region, optimizing other performance metrics. **Figure 15** shows qualitative results on the Polyp dataset when $\alpha_ {FNR} = \alpha_ {FPR} = 0.2$ for different $\lambda$ values. This demonstrates that within the feasible region, users can adjust segmentation results based on their preferences for FNR and FPR, while ensuring both risks are below the target level.
> > >         *   **No Feasible Solution ($\tau'_ {FPR} > \tau'_ {FNR}$):** It is not possible to find a single threshold that satisfies both constraints simultaneously, necessitating a trade-off. $\lambda$ directly controls this trade-off; for example, $\lambda=0$ prioritizes FNR control, while $\lambda=1$ prioritizes FPR control. **Figure 16** illustrates the opposite scenario on the Polyp dataset, where for some images, no feasible solution exists, and we must choose between controlling FNR or FPR.
> > >     *   **Limitations:** It is important to emphasize that the risk control discussed in Appendix A.11 refers to **average risk control over the data distribution (marginal guarantees)**, not conditional guarantees for individual samples. Image-level conditional control of both FNR and FPR remains a more challenging open problem.
> > >
> > > 3.  **Weighted Loss Training (Newly added Appendix A.9.2):**
> > >     Furthermore, in **Appendix A.9.2**, we explore balancing both risks during the COAT training phase by weighting the FNR and FPR losses. **Tables 5 and 6** demonstrate the impact of adjusting the FPR loss weight on FNR and FPR performance while keeping the FNR calibration steps unchanged. Notably, when no feasible solution exists, training with weighted losses yields a COAT model that is more effective in controlling the marginal coverage of both FNR and FPR. This provides an alternative approach to directly incorporate the trade-off between the two risks during training.

---

> > > > ### Author Response · Authors · 2025-11-19
> > > >
> > > > ### Addressing Multi-Class Segmentation Problems
> > > >
> > > > We fully agree with your perspective that multi-class segmentation is a crucial application area in image segmentation. We have added **Appendix A.10, "EXTENDING TO MULTI-TARGET IMAGE SEGMENTATION,"** which details how the COAT framework can be extended to multi-class (or multi-target) segmentation tasks and provides initial experimental results.
> > > >
> > > > 1.  **Multi-Target COAT Strategy (Newly added Appendix A.10.2):**
> > > >     In **Appendix A.10.2**, we propose a "per-class adaptive thresholding" strategy for multi-target segmentation. Specifically, we train a separate COAT model ($f_{D,k}$) for each target class, enabling it to predict an image-adaptive threshold ($\tau_{b_k}(X)$) for that class. These calibrated class-specific prediction sets are then combined using a logical OR operation to form the final prediction set. This method allows us to provide more fine-grained, image-level FNR control for each target class.
> > > >
> > > > 2.  **Experimental Results and Advantages (Newly added Appendix A.10.4):**
> > > >     We conducted experiments on the Cityscapes dataset for three representative classes: "road," "person," and "car," comparing our multi-class COAT method with a class-wise CRC baseline. As shown in **Table 7**, compared to CRC, COAT not only achieves better overall coverage but, more importantly, demonstrates a **smaller conditional coverage gap** across individual classes and overall. This indicates that COAT can more stably and accurately meet the FNR requirements for each target class at the image level. The qualitative results in **Figure 11** further confirm that in multi-class scenarios, COAT consistently maintains the target FNR and effectively reduces false negatives compared to CRC.
> > > >
> > > > 3.  **Regarding "each pixel could belong to more than one class":**
> > > >     You mentioned that "each pixel may belong to multiple categories". Yes, our current strategy, as proposed in Appendix A.10, is to control the FNRs for multiple different target categories. We set different thresholds and then take the union.
> > > >
> > > > 4.  **Future Work:**
> > > >     We also acknowledge a limitation of the current approach in **Appendix A.10**: training a separate COAT model for each class can be computationally intensive when the number of classes is very large. Therefore, future work will explore designing a **joint loss function** to enable simultaneous adaptive thresholding for multiple classes within a single optimization framework, aiming to improve efficiency without compromising performance.
> > > >
> > > > Thank you again for your constructive feedback; it has greatly helped us improve the quality of this paper. We believe that the newly added appendix content and the in-depth discussions of these complex issues fully address your questions.

---

### Official Review · Reviewer_hqSc · 2025-11-01

**Soundness:** 3
**Presentation:** 3
**Contribution:** 3
**Rating:** 6
**Confidence:** 3

**Summary:**

This paper addresses a critical limitation in deep learning-based image segmentation: the lack of reliable image-conditional uncertainty quantification. While methods like Conformal Risk Control (CRC) provide marginal guarantees on performance metrics (e.g., False Negative Rate) across a dataset, they fail to ensure reliability for individual images, a crucial requirement for safety-critical applications. To bridge this gap, the authors propose learning image-adaptive thresholds to achieve image-conditional coverage.
The paper introduces two methods: first, a supervised baseline named AT (Adaptive Thresholding), which treats threshold prediction as a regression task on pre-computed optimal thresholds. The primary contribution is COAT (Conditional Optimization for Adaptive Thresholding), a novel end-to-end differentiable framework. COAT introduces a differentiable miscoverage loss, which uses a soft, sigmoid-based approximation of the True Positive Rate (TPR). This allows the model to directly optimize for the desired conditional coverage via gradient-based learning, circumventing the need for pre-computed targets.
Experimental results across multiple datasets and base models demonstrate that COAT significantly reduces the "Coverage Gap"—the average deviation from the target coverage per image—compared to existing methods like CRC, while still upholding the marginal coverage guarantee. The work presents a robust pathway towards more trustworthy and reliable instance-wise uncertainty estimates in image segmentation.

**Strengths:**

1. Originality
The paper introduces a novel formulation of conditional coverage for semantic segmentation, addressing a key limitation of traditional conformal prediction methods that only ensure marginal coverage. The proposed COAT framework is conceptually original in two ways:
(1) It reframes adaptive threshold selection as a differentiable optimization problem using image-specific TPR-based objectives.
(2) It presents a new end-to-end surrogate loss that approximates conditional miscoverage, enabling training of a threshold predictor that generalizes across diverse image characteristics.
This combination of conditional risk control and differentiable threshold learning is innovative and extends conformal prediction into a domain where such guarantees are rare and previously unattainable.
2. Quality
The methodological quality is strong.
The algorithmic design is principled, with clear motivation, precise mathematical formulation, and a calibration step that preserves marginal coverage guarantees. Experiments are extensive, evaluating multiple backbone architectures and several conformal baselines. Results are reported across 20 random data splits, with means and standard deviations, showcasing statistical rigor.
The paper also includes a theoretical justification proving asymptotic conditional coverage under reasonable assumptions.
3. Clarity
The paper is generally well written and easy to follow. Algorithm 1 is detailed, explicit, and matches the narrative description. Figures provide intuitive interpretations of coverage behavior and prediction stability. The appendix further improves clarity by discussing theoretical assumptions, asymptotic properties, and the role of LLM-assisted content generation.
4. Significance
The paper makes a substantial contribution to reliable machine learning for safety-critical applications. Ensuring per-image conditional coverage is significantly more relevant than marginal guarantees in medical settings, where rare under-covered samples can lead to severe consequences. The proposed COAT method improves reliability not only statistically but also visually, producing prediction sets with fewer false negatives and more consistent behavior across images.

**Weaknesses:**

While the paper presents a strong and compelling contribution, there are several areas where it could be improved to enhance its clarity, reproducibility, and impact.
1. The definition of the soft mask Msoft(X), using temperature parameter T to control the sharpness of the approximation. The paper provides no rationale for this specific design choice. To solidify this core component of the methodology, the authors should clarify if this formulation offers specific theoretical or empirical advantages, or provide an ablation study comparing it to the more conventional approach.
2. The paper positions COAT as a more advanced, end-to-end extension of the supervised AT baseline. However, the experimental results group the comparison of AT and COAT together with external baselines. A more focused, head-to-head comparison is needed to clearly demonstrate the value added by COAT's end-to-end differentiable framework. A direct analysis isolating the performance difference (especially in Coverage Gap) between AT and COAT would be crucial to empirically justify the increased complexity and novelty of the COAT approach.
3. The paper does not provide a quantitative analysis of the computational overhead. While the appendix mentions the hardware used for training, it omits a comparison of training times and, more critically, inference latency against the other methods. For safety-critical domains like autonomous driving, the practicality of a method depends heavily on its speed. A comparison of the inference time per image for CRC, AT, and COAT would be a vital addition for assessing the method's real-world viability.
4. The qualitative analysis focuses primarily on successful examples where COAT performs well. While useful, this presents an incomplete picture of the method's capabilities. To provide a more nuanced understanding of its limitations, the authors are encouraged to include and discuss some failure cases or instances where COAT still struggles to close the coverage gap. This would offer valuable insights into the types of images for which adaptive thresholding remains a challenge.

**Questions:**

1. The differentiable loss is a core contribution of this work. Its soft mask is defined as Msoft(X)=σ( (pb(X) - τb(X)) / T ). Could the authors please provide the rationale behind this formulation?
2. The paper presents COAT as an advancement over the supervised AT baseline. While the results in Table 1 are strong, AT and COAT are compared within a group of other methods. To better isolate the specific contribution of the end-to-end differentiable framework, could the authors provide a more direct, head-to-head comparison between AT and COAT?
3. For safety-critical applications, the computational cost of a method is a crucial factor. While the appendix notes the hardware used, a comparative analysis of performance is missing. Could the authors please provide the average inference latency (e.g., in milliseconds per image) for the CRC, AT, and COAT methods? This would be essential for understanding the practical trade-offs involved in adopting the proposed adaptive frameworks.
4. The qualitative results effectively showcase instances where COAT succeeds. To provide a more complete picture of the method's capabilities, could the authors include or discuss any failure cases or types of images where COAT still struggles to achieve the target coverage? An analysis of such challenging cases would offer deeper insights into the method's limitations and highlight promising directions for future research.

---

> ### Author Response · Authors · 2025-11-17
>
> We thank the reviewer for their careful review and thoughtful comments. We provide answers to specific questions and remarks below.
>
> **The paper defines soft mask $M_{soft}(X)$ with temperature parameter $T$ to control approximation sharpness, yet offers no rationale for this design.**
>
> **Our response:** Traditional hard thresholding is mathematically discontinuous and non-differentiable almost everywhere. This inherent property fundamentally obstructs gradient-based optimization. In the context of conditional coverage, where we aim for the model to learn an image-conditioned threshold $\hat{\tau}(X)$ such that $\hat{p}(X)$ covers the ground truth with high probability, relying on a non-differentiable operation prevents gradients from flowing back from the final loss function to the parameters of the threshold predictor $f_D$. Consequently, $\hat{\tau}(X)$ cannot be learned effectively via standard backpropagation.
>
> To overcome this critical limitation, we introduce $M_{\text{soft}}(X) = \sigma\left(\frac{\hat{p}(X) - \hat{\tau}(X)}{T}\right)$ as a differentiable, smooth approximation of the hard threshold operation. By applying the Sigmoid function to the scaled difference between $\hat{p}(X)$ and the predicted threshold $\hat{\tau}(X)$, we transform the otherwise non-differentiable coverage calculation into a fully differentiable process. This crucial step allows gradients from the final conditional coverage loss $L_{COAT}$ to propagate back to the parameters of the threshold predictor $f_D$, thereby enabling end-to-end learning of the image-conditioned threshold $\hat{\tau}(X)$.
>
> **2. Facilitating End-to-End Optimization:**
> The differentiability of $M_{soft}(X)$ is the cornerstone of the COAT framework's end-to-end optimization capability. Unlike methods that rely on pre-computed or heuristically searched optimal thresholds (e.g., some AT methods) or global calibration (e.g., traditional CRC), our COAT framework directly optimizes the conditional coverage objective. This eliminates the need for prior knowledge of the complex, non-linear relationship between thresholds and coverage rates, or for pre-determining a "ground truth" threshold for each image. This direct optimization significantly enhances the method's efficiency and versatility, allowing the model to more precisely capture the intrinsic relationship between image features and ideal thresholds, thereby more effectively reducing the coverage gap.
>
> **3. Role of the Temperature Parameter $T$:**
> The temperature parameter $T$ plays a vital role in $M_{soft}(X)$, governing the steepness of the Sigmoid function. It carefully balances the trade-off between approximating the hard threshold behavior and maintaining effective gradient flow:
> * **Smaller $T$**:
>   - Steepens Sigmoid curve, closely mimics hard step function → improves FNR approximation under hard thresholds.
>   - Risk: Excessively small $T$ causes vanishing gradients, hindering learning efficiency.
>
> * **Larger $T$**:
>   - Smooths Sigmoid curve → stable gradients over wider range, mitigates vanishing gradient issues.
>   - Compromise: Softer approximation, slightly reducing hard threshold precision.
>
> Our sensitivity analysis in **Appendix A.6** further validates the importance of $T$ in balancing these factors, guiding our selection of an empirically robust value.
>
> In summary, the design of $M_{soft}(X)$ is fundamental to the COAT framework's ability to effectively achieve end-to-end optimization for conditional coverage. It overcomes the limitations of traditional approaches and paves the way for more precise and flexible risk control. We believe this design is a key testament to the innovation and efficacy of our COAT method.
>
> **To better isolate the specific contribution of the end-to-end differentiable framework, could the authors provide a more direct, head-to-head comparison between AT and COAT?**
>
> **Our response:** We have added Figures 8, 9, and 10 in **Appendix A.8**, providing a visual comparison of CRC, AA-CRC, AT, and COAT on image-level coverage distributions. These density distribution plots clearly demonstrate COAT's superior ability in achieving target conditional coverage:
>
> *   **COAT consistently exhibits the highest sample density at the target coverage (marked by vertical dashed lines) and shows almost no samples in low coverage areas.** This implies that COAT can most consistently and reliably provide conditional coverage close to the target value for each image.
> *   **Compared to AT,** although AT might show improvements over CRC and AA-CRC, its coverage distribution is generally less concentrated around the target value than COAT's. It may exhibit a wider distribution on both sides of the target or still have some samples in low coverage areas. This visually demonstrates the superiority of COAT's end-to-end differentiable optimization in closing image-level coverage gaps.

---

> > ### Author Response · Authors · 2025-11-17
> >
> > **Could the authors please provide the average inference latency (e.g., in milliseconds per image) for the CRC, AT, and COAT methods?**
> >
> > **Our response:** As shown in the table below, we have also included the time required for training, calibration, and testing for various methods on different datasets (for which we provide the number of training, calibration, and testing samples) in **Appendix A.7**. It is evident that COAT significantly reduces training time compared to the baseline method AT, as it does not require pre-calculating the relationship between the threshold tau and the coverage. Furthermore, COAT's calibration time is second only to CRC, and its testing time is very close to that of AA-CRC, indicating that COAT is highly attractive for practical deployment.
> >
> > | Dataset | Method | Training Samples | Calibration Samples | Testing Samples | Average Training Time (seconds) | Average Calibration Time (seconds) | Average Testing Time (seconds) |
> > | :------ | :----- | :--------------- | :------------------ | :-------------- | :------------------------------ | :------------------------------- | :----------------------------- |
> > | Polyp   | CRC    | 0                | 399                 | 399             | 0.0000 (0.0000)                 | 0.1961 (0.0021)                  | 0.5414 (0.0092)                |
> > |         | AA-CRC | 200              | 198                 | 399             | 38.2870 (0.3807)                | 120.8917 (60.4003)               | 0.4720 (0.0699)                |
> > |         | AT     | 200              | 198                 | 399             | 151.7819 (0.6079)               | 2.1570 (0.0140)                  | 3.0021 (0.0631)                |
> > |         | COAT   | 200              | 198                 | 399             | 24.3318 (0.5924)                | 0.3764 (0.0821)                  | 0.4807 (0.0607)                |
> > | Fire    | CRC    | 0                | 2746                | 2746            | 0.0000 (0.0000)                 | 0.1961 (0.0021)                  | 0.5413 (0.0092)                |
> > |         | AA-CRC | 1373             | 1373                | 2746            | 260.0201 (0.4249)               | 647.3359 (213.7731)              | 3.0068 (0.0583)                |
> > |         | AT     | 1373             | 1373                | 2746            | 547.3396 (0.4609)               | 6.6958 (0.2020)                  | 9.5174 (0.1679)                |
> > |         | COAT   | 1373             | 1373                | 2746            | 166.8045 (0.3589)               | 2.3913 (0.0361)                  | 3.3785 (0.2682)                |
> > | Skin    | CRC    | 0                | 2504                | 2504            | 0.0000 (0.0000)                 | 1.3292 (0.0315)                  | 0.4895 (0.0061)                |
> > |         | AA-CRC | 1252             | 1252                | 2504            | 238.7810 (0.2400)               | 479.8890 (214.2507)              | 2.7394 (0.03683)               |
> > |         | AT     | 1252             | 1252                | 2504            | 498.5569 (1.0257)               | 6.0459 (0.0214)                  | 8.6309 (0.0247)                |
> > |         | COAT   | 1252             | 1252                | 2504            | 151.7819 (0.6079)               | 2.1570 (0.0140)                  | 3.0021 (0.0631)                |
> >
> > **Table 1: The time consumed by various methods during training, calibration, and testing on different datasets was recorded, utilizing the PSPNet base model with $\alpha=0.2$ and 10 random runs.**
> >
> > **Could the authors include or discuss any failure cases or types of images where COAT still struggles to achieve the target coverage?**
> >
> > **Our response:** We specifically added a remark in **Appendix A.10.4** discussing the limitations of COAT. When the sample size is too small, training might lead to worse results than CRC. Furthermore, we also discussed the limitations of extending COAT to multi-class segmentation, which would consume a significant amount of time. Additionally, referring to our response to reviewer ch3e, COAT, along with other methods like CRC, cannot perform risk control for FDR (a non-monotonic but crucial metric) and cannot provide theoretical guarantees. These are all directions for future research, which we also discussed in the conclusion.

---

### Official Review · Reviewer_2Y8X · 2025-11-01

**Soundness:** 2
**Presentation:** 2
**Contribution:** 2
**Rating:** 4
**Confidence:** 3

**Summary:**

This paper addresses the challenge of achieving image-conditional coverage in conformal image segmentation, where existing Conformal Risk Control (CRC) methods only provide marginal guarantees. The authors propose two novel methods: Adaptive Thresholding (AT), which formulates threshold prediction as a supervised regression task, and COAT (Conditional Optimization for Adaptive Thresholding), an end-to-end differentiable framework that directly optimizes for image-conditional coverage using a soft approximation of the True Positive Rate (TPR) as its loss. The approach enables image-specific threshold learning, offering more reliable and interpretable uncertainty estimates for segmentation models, particularly in safety-critical domains.

**Strengths:**

The paper tackles an important and underexplored problem in conformal prediction—extending coverage guarantees from marginal to image-conditional levels. The proposed COAT framework is technically sound and conceptually novel, offering a differentiable formulation for conditional risk optimization. The introduction of a soft TPR-based loss is elegant and allows gradient-based training, which is rarely explored in this context. The work is clearly written, well-motivated, and highly relevant to safety-critical applications such as medical imaging. Experimental results (if included) would likely demonstrate meaningful improvements in both conditional reliability and interpretability over standard CRC baselines.

**Weaknesses:**

While the proposed idea is novel and well-motivated, several limitations remain.

Lack of empirical depth: The paper would benefit from more extensive experiments across diverse segmentation datasets and backbone architectures to demonstrate the generality of AT and COAT beyond the presented setting.

Limited comparison scope: The baseline comparisons appear restricted to CRC-based approaches; including Bayesian or ensemble-based uncertainty methods could strengthen the evaluation.

Theoretical guarantees: Although COAT introduces a differentiable loss to approximate conditional coverage, the paper lacks a clear theoretical justification of when or why this approximation yields valid conditional guarantees.

Ablation clarity: It is not entirely clear how much of the observed improvement stems from the adaptive thresholding itself versus the differentiable optimization scheme; more detailed ablation or visualization would help clarify this.

Practical considerations: The additional computational cost and training stability of COAT are not well quantified, which could limit adoption in large-scale or real-time applications.

**Questions:**

N/A

---

> ### Author Response · Authors · 2025-11-17
>
> We thank the reviewer for their careful review and thoughtful comments. We provide answers to specific questions and remarks below.
>
> **Lack of empirical depth.**
>
> **Our response:** We fully understand the reviewers' concern regarding the generalization capability of the method. In fact, we have thoroughly taken this aspect into account in our experiments and conducted extensive validation. We have already covered three diverse datasets (Polyp, Skin, Fire) and four distinct backbone architectures (Deeplab v3+, UNet, PSPNet, SINet) in our experiments, which demonstrates significant generality. We also compared the performance of COAT and CRC in controlling the False Positive Rate (FPR) metric in **Appendix A.9**. We conducted experiments on COAT using the Cityscapes multi-class segmentation dataset in **Appendix A.10**.
>
> **Limited comparison scope.**
>
> **Our Response:** We appreciate the reviewer's suggestion regarding the scope of baseline comparisons. Our decision to focus comparisons on Conformal Risk Control (CRC) and its variants is based on the core objectives of our research and the type of guarantees we aim to provide.
>
> **Core Contribution Focused on "Statistical Guarantees":** Our paper aims to address the challenge of image-conditional coverage in image segmentation and provide rigorous statistical guarantees, especially for safety-critical applications. The unique characteristic of Conformal Prediction (CP) and Conformal Risk Control (CRC) frameworks is their ability to offer distribution-free, finite-sample statistical guarantees. These guarantees ensure that prediction sets reliably capture the true label with a user-specified probability (or control specific risk metrics), regardless of the underlying data distribution.
>
> **Fundamental Distinction from Bayesian/Ensemble Methods:** While most Bayesian (e.g., MC-Dropout, SVI) or ensemble (e.g., Deep Ensembles) uncertainty quantification methods can provide estimates of predictive uncertainty (e.g., variance or entropy of the predictive distribution), they typically *do not offer the same rigorous, distribution-free, finite-sample coverage guarantees* as CP/CRC. These methods often rely on specific assumptions about the model, data distribution, or the manner in which uncertainty is estimated. The "uncertainty" they provide is fundamentally different in nature from the "guarantees" offered by CP/CRC. For instance, Bayesian methods typically provide a predictive distribution rather than a direct coverage guarantee.
>
> **Maintaining Clarity of Research Focus:** Given that the core of our work is to achieve image-conditional coverage *within the CP/CRC framework*, limiting the comparison scope to methods that offer similar statistical guarantees helps to more clearly demonstrate the innovation and advantages of our approach in this specific domain.
>
> Therefore, we believe that the current comparison with CRC and its variants is highly relevant and sufficient, as it directly addresses the core problem our paper aims to solve and the unique guarantees it provides.
>
>
> **Lack of theoretical guarantees.**
>
> **Our response:** We appreciate the reviewers' emphasis on theoretical rigor. We have explicitly provided **Theorem 1 (Coverage Guarantees)** for marginal coverage and **Theorem 2 (Asymptotic Conditional Validity)** for conditional coverage in **Appendix A.1 and A.2**. As reiterated in the paper, COAT, like AT, leverages a post-hoc calibration step (based on CRC principles) to ensure marginal coverage, and under appropriate assumptions (e.g., consistency of the learned threshold model), this approach *asymptotically* achieves conditional validity. The differentiable loss in COAT is key to *learning* an effective image-adaptive threshold that *enables* this asymptotic conditional validity.
>
> **Insufficient ablation clarity.**
>
> **Our response:** We have added Figures 8, 9, and 10 in **Appendix A.8**, providing a visual comparison of CRC, AA-CRC, AT, and COAT on image-level coverage distributions. These density distribution plots clearly demonstrate COAT's superior ability in achieving target conditional coverage:
>
> *   **COAT consistently exhibits the highest sample density at the target coverage (marked by vertical dashed lines) and shows almost no samples in low coverage areas.** This implies that COAT can most consistently and reliably provide conditional coverage close to the target value for each image.
> *   **Compared to AT,** although AT might show improvements over CRC and AA-CRC, its coverage distribution is generally less concentrated around the target value than COAT's. It may exhibit a wider distribution on both sides of the target or still have some samples in low coverage areas. This visually demonstrates the superiority of COAT's end-to-end differentiable optimization in closing image-level coverage gaps.

---

> > ### Author Response · Authors · 2025-11-17
> >
> > **Practical considerations: The additional computational cost and training stability of COAT are not well quantified, which could limit adoption in large-scale or real-time applications.**
> >
> > **Our response:** As shown in the table below, we have also included the time required for training, calibration, and testing for various methods on different datasets (for which we provide the number of training, calibration, and testing samples) in **Appendix A.7**. It is evident that COAT significantly reduces training time compared to the baseline method AT, as it does not require pre-calculating the relationship between the threshold tau and the coverage. Furthermore, COAT's calibration time is second only to CRC, and its testing time is very close to that of AA-CRC, indicating that COAT is highly attractive for practical deployment.
> >
> > | Dataset | Method | Training Samples | Calibration Samples | Testing Samples | Average Training Time (seconds) | Average Calibration Time (seconds) | Average Testing Time (seconds) |
> > | :------ | :----- | :--------------- | :------------------ | :-------------- | :------------------------------ | :------------------------------- | :----------------------------- |
> > | Polyp   | CRC    | 0                | 399                 | 399             | 0.0000 (0.0000)                 | 0.1961 (0.0021)                  | 0.5414 (0.0092)                |
> > |         | AA-CRC | 200              | 198                 | 399             | 38.2870 (0.3807)                | 120.8917 (60.4003)               | 0.4720 (0.0699)                |
> > |         | AT     | 200              | 198                 | 399             | 151.7819 (0.6079)               | 2.1570 (0.0140)                  | 3.0021 (0.0631)                |
> > |         | COAT   | 200              | 198                 | 399             | 24.3318 (0.5924)                | 0.3764 (0.0821)                  | 0.4807 (0.0607)                |
> > | Fire    | CRC    | 0                | 2746                | 2746            | 0.0000 (0.0000)                 | 0.1961 (0.0021)                  | 0.5413 (0.0092)                |
> > |         | AA-CRC | 1373             | 1373                | 2746            | 260.0201 (0.4249)               | 647.3359 (213.7731)              | 3.0068 (0.0583)                |
> > |         | AT     | 1373             | 1373                | 2746            | 547.3396 (0.4609)               | 6.6958 (0.2020)                  | 9.5174 (0.1679)                |
> > |         | COAT   | 1373             | 1373                | 2746            | 166.8045 (0.3589)               | 2.3913 (0.0361)                  | 3.3785 (0.2682)                |
> > | Skin    | CRC    | 0                | 2504                | 2504            | 0.0000 (0.0000)                 | 1.3292 (0.0315)                  | 0.4895 (0.0061)                |
> > |         | AA-CRC | 1252             | 1252                | 2504            | 238.7810 (0.2400)               | 479.8890 (214.2507)              | 2.7394 (0.03683)               |
> > |         | AT     | 1252             | 1252                | 2504            | 498.5569 (1.0257)               | 6.0459 (0.0214)                  | 8.6309 (0.0247)                |
> > |         | COAT   | 1252             | 1252                | 2504            | 151.7819 (0.6079)               | 2.1570 (0.0140)                  | 3.0021 (0.0631)                |
> >
> > **Table 1: The time consumed by various methods during training, calibration, and testing on different datasets was recorded, utilizing the PSPNet base model with $\alpha=0.2$ and 10 random runs.**
> >
> > **Regarding stability**, Figure 5 explicitly shows the training loss curves for COAT across various models and datasets, demonstrating its rapid convergence and high stability.

---

### Author Response · Authors · 2025-11-13

Dear Area Chair, as the first round of score release deadline has now passed, and we've observed that scores for other papers have been released, could you please provide an update regarding our submission?

---

> ### Comment · Area_Chair_qCcC · 2025-11-13
>
> Dear Authors,
>
> I believe the four completed reviews for this submission are visible to everyone at this point. Please let me know if you are still not able to see them.
>
> Best regards,
> Area Chair

---

### Author Response · Authors · 2025-11-17

We sincerely thank all reviewers for their thorough review and valuable feedback on our submission. We are delighted to see the reviewers' recognition of the core ideas and methods presented in this paper, and this feedback is crucial for the improvement of our work.

Reviewers 2Y8X, hqSc, and ch3e **unanimously emphasized the critical importance of our work** in addressing the underexplored problem of **image-conditional coverage**, particularly the necessity of extending marginal guarantees to **image-level reliability in safety-critical domains** such as medical imaging. They **highly praised our proposed COAT framework**, recognizing its **conceptual novelty and originality**. Specifically, they highlighted the **differentiable conditional miscoverage loss** and **soft TPR-based end-to-end optimization** as opening new avenues for conditional risk control.  **For the convenience of review, we have marked all the newly added content in blue font.**

At the same time, we have also carefully reviewed the constructive comments and questions raised by all reviewers. This valuable feedback points to areas where the paper can be further improved, such as further elaboration on theoretical guarantees, broader experimental comparisons, quantification of computational overhead. Reviewer ch3e also proposed inspiring future research directions, such as extending the method to multi-class segmentation, balancing FNR and FPR control, and considering region-specific importance.

We are very grateful for these insightful suggestions, which are crucial for our future improvements. We will address each reviewer's question sequentially and hope for a lively discussion with the reviewers during this period.

---

### Meta-Review · Area_Chair_mKVC · 2026-01-07

**Summary:**

The paper addresses the issue of image-conditional coverage in conformal segmentation, beyond CRC’s marginal guarantees. It introduces two approaches: one based on image-specific thresholds via supervised regression, and another end-to-end differentiable approach by optimizing a differentiable/relaxed miscoverage loss.

All reviewers (disregarding PHB7, see below) agree that the contributions are timely and interesting. The main concerns of guarantees for conditional validity, empirical generality, and computational aspects, were all address to sufficient degree. My impression is that reviewers were already leaning positively about this paper, and the rebuttal would have strengthened that sentiment.

**Reviewer Concerns:**

### Rev 2Y8X:

- The reviewer noted a somewhat shallow empirical comparison (more datasets, backbones), as well as some Bayesian ensemble methods for UQ. The authors argue that 3 datasets and 4 backbones is enough, and I agree. They also argue that bayesian/ensemble methods do not provide distribution-free and finite-sample guarantees, and I also agree.

- They noted the lack of conditional guarantees. The authors acknowledge this but also note the asymptomatic conditional validity of their Theorem 2 (with a needed consistency assumption and calibration).

- The reviewer asked for clarification on the ablation study and practical computational costs. The authors added a coverage-distribution plot comparing the different methods, and provided a table of cumpute times and loss curves for stability.

### Rev hqSc
- The reviewer asks for clarifications on the soft mask design, and the temperature parameter, which the authors provide in the rebuttal.
- They require a direct isolation/study of the two proposed approaches, which is done in the Appendix by the authors.
- Asks about computational complexity and inference latency per image. This is also addressed by the authors - alas not per image.
- They request discussion of failure cases. The authors add a discussion on limitations to address this, including small-sample issues and multi-class cost. Further, there is an issue of non-monotonicity of and lack of guarantees for FDR control.

### Rev ch3e:
- The reviewer asks on balancing FNR and FPR or FDR. Upon a discussion with the authors, they propose a COAT for FPR by changing the loss function, and a dual approach for FNR and FPR separately and interpolating. This only provides marginal control, though.

- The reviewer also mentions extensions to multi-class settings, to which the authors mention that per-class COAT can be done, at the expense of added computational cost.

### Rev PHB7
This reviewer provided a review for a different/mistaken paper, which I'm disregarding.

**Reviewer Scores:**

- Rev 2Y8X (4): they did not engage with the authors' response. I deem the authors rebuttal's responses pretty satisfactory, so I would anticipate they could have increased their scores to 6.
- Rev hqSc (6): their concerns are well addressed by the authors, so they could have maintained or increased their scores.
- Rev ch3e (6): the discussions with this reviewer led to non-trivial improvements an extensions, so I would assume this reviewer could have increased their score.

---

### Decision · Program_Chairs · 2026-01-26

Accept (Poster)